

# DEUCE v1.0: A neural network for probabilistic precipitation nowcasting with aleatoric and epistemic uncertainties

Bent Harnist[1], Seppo Pulkkinen[1], and Terhi Mäkinen[1]

[1]Finnish Meteorological Institute, Erik Palménin aukio 1, FI-00560 Helsinki, Finland

**Correspondence:** Bent Harnist (bent.harnist@fmi.fi)

**Abstract.**

Precipitation nowcasting (forecasting locally for 0–6h) serves both public security and industries, facilitating the mitigation of losses incurred due to e.g. flash floods, and is usually done by predicting weather radar echoes, which provides better performance than NWP at that scale. Probabilistic nowcasts are especially useful as they provide a desirable framework for

operational decision-making. Many extrapolation-based statistical nowcasting methods exist, but they all suffer from a limited ability to capture the nonlinear growth and decay of precipitation, leading to a recent paradigm shift towards deep learning methods, more capable of representing these patterns.

Despite of its potential advantages, the application of deep learning in probabilistic nowcasting has only recently started to be explored. Here we develop a novel probabilistic precipitation nowcasting method, based on Bayesian neural networks

with variational inference and the U-Net architecture, named DEUCE. The method estimates the total predictive uncertainty of precipitation by combining estimates of the epistemic (knowledge-related, reducible) and heteroscedastic aleatoric (data-dependent, irreducible) uncertainties, and produces an ensemble of development scenarios for the following 60 minutes.

DEUCE is trained and verified using Finnish Meteorological Institute radar composites against established classical models. Our model is found to produce both skillful and reliable probabilistic nowcasts based on various evaluation criteria. It improves

ROC Area Under the Curve scores 1–5% over STEPS and LINDA-P baselines, and comes close to the best-performer STEPS on a CRPS metric. The reliability of DEUCE is demonstrated with, e.g., having the lowest Expected Calibration Error at 20 and 25 dBZ reflectivity thresholds, and coming second at 35 dBZ. On the other hand, deterministic performance of ensemble means is found to be worse than that of extrapolation and LINDA-D baselines. Lastly, the composition of the predictive uncertainty is analysed and described, with the conclusion that aleatoric uncertainty is more significant and informative than epistemic

uncertainty in the DEUCE model.





## DEUCE v1.0: A neural network for probabilistic precipitation nowcasting with aleatoric and epistemic uncertainties

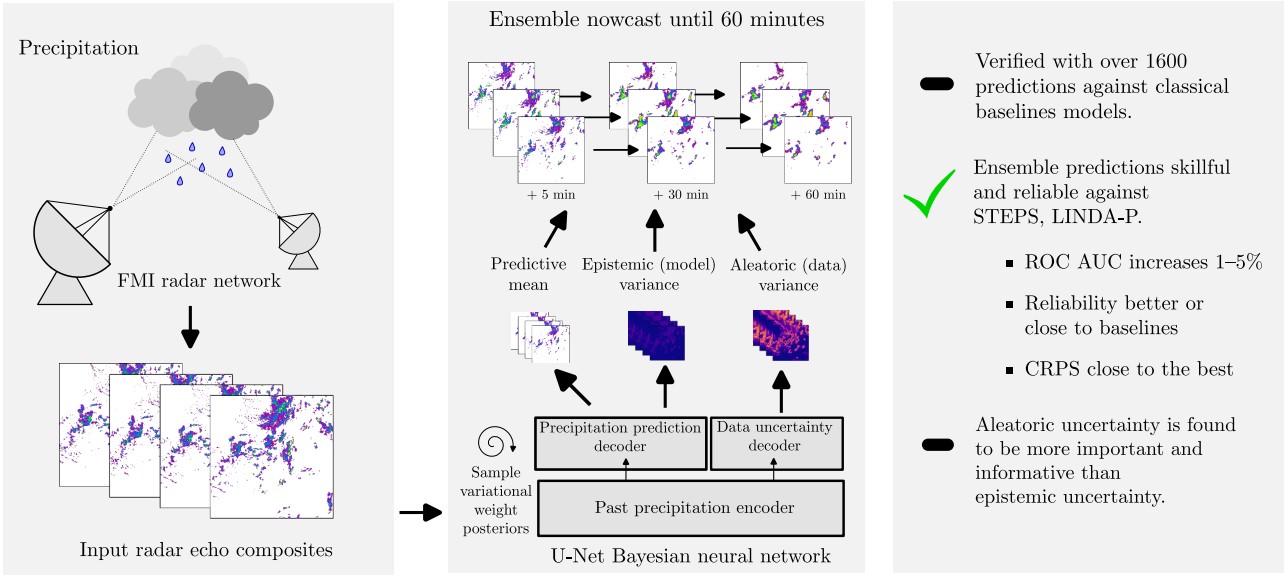

## 1 Introduction

Predicting the amount and location of precipitation at local scales of a few kilometres for lead times ranging from minutes to hours, i.e., *precipitation nowcasting* has recently grown into an important component of severe weather early warning systems, particularly those focused on predicting flash floods. Because of the intensification coupled with increased frequency of extreme precipitation events brought by climate change, accurate estimates of future precipitation have increased in importance. However, the capacity of any nowcasting model to produce accurate estimates is limited, and thus having additionally an idea of the reliability of the nowcast is operationally important. This can be addressed with ensemble nowcasts, which generate a set of possible scenarios, with which it is possible to estimate the probability of certain events.

Numerical Weather Prediction (NWP) is widely used for forecasts at longer timescales and with coarser grids (Bauer et al., 2015), with regional high-resolution models model generally having a grid resolution of a few kilometres and a refresh rate of typically one hour. For example, the High Resolution Rapid Refresh (HRRR) model developed by the United States National Oceanic and Atmospheric Administration (NOAA) has a grid resolution of 3km and a refresh rate of one hour (Alexander et al., 2020). However, NWP does not achieve sufficient performance at the spatio-temporal scales typical of nowcasts, due to not yet having achieved numerical stability in these first few hours, and due to the computational complexity of resolving atmospheric equations at sub-hour temporal resolutions and grid resolutions approaching the micro-scale ($\leq$ 1km) (Sun et al., 2014; Radhakrishnan and Chandrasekar, 2020). Specialized nowcasting methods for precipitation have been developed in parallel of NWP and may be used in order to circumvent its problems in the domain. These mainly rely on forecasting the





evolution of radar echo image sequences, that act as a good proxy for ground-level precipitation, and usually have a spatial
resolution of ∼ 1 km and a temporal resolution of ∼ 5 min, which are characteristic of weather radar observations.

## 1.1 Extrapolation-based precipitation nowcasting

The most important class of precipitation nowcasting models is based on the extrapolation of radar echoes along the background
advection field. These models first estimate the advection field from a sequence of past radar images, with methods such as
Variational Echo Tracking (Laroche and Zawadzki, 1995) or optical flow-based methods like the Lucas-Kanade method (Lucas
and Kanade, 1981; Bouguet et al., 2001). In the classical case of the pure extrapolation nowcast, the most recently observed
frame is simply extrapolated along the estimated advection field, often using a Semi-Lagrangian Scheme (Staniforth and Côté,
1991). Extrapolation nowcasting doesn't model the growth and decay of precipitation, so many extensions attempting to make
up for that have been developed. One important method is Spectral Prognosis (S-PROG) by Seed (2003). S-PROG is based on
the scale-dependence of the lifetime and evolution of features, decomposing the field into additive components corresponding
to different spatial scales and evolving each of them separately using an autoregressive (AR2) model in Lagrangian (flow frame
of reference) coordinates, enabling modeling the scale-dependent behavior of precipitation.

STEPS (Short-Term Ensemble Prediction System) by Bowler et al. (2006), is an influential ensemble nowcasting model
based on S-PROG. In STEPS, stochastic perturbations are added to the motion field in order to model its uncertainty. Just like
with S-PROG, the growth and decay of the precipitation field is modeled by decomposing it using a cascade of scales with the
autoregressive model applied to each of these scales separately in Lagrangian coordinates. Unlike in S-PROG, stochastic noise
is injected at each scale, concurrently with the AR modeling. Over time, various models have expanded upon STEPS; one recent
example being LINDA (Lagrangian INtegro-Difference equation model with autoregression) (Pulkkinen et al., 2021), which
uses an integro-difference equation model with rain cell detection and convolutions for modeling the loss of predictability at
small scales. LINDA produces nowcasts particularly well-suited to the prediction of strong localized rainfall.

## 1.2 Deep Learning approaches to precipitation nowcasting

With significant recent advances in deep learning, the interest in its use for precipitation nowcasting has increased. One of the
first deep learning model to have been used explicitly for precipitation nowcasting is the Convolutional LSTM (ConvLSTM)
model (Shi et al., 2015), which combines the temporal prediction capacity of the Long Short-Term Memory (LSTM) neural
networks with 3D convolutions modeling spatiotemporal features in one model for spatiotemporal nowcasting. ConvLSTM
has later been improved by the TrajGRU model (Shi et al., 2017), that replaces the heavy LSTM structure with a lighter GRU
(Gated Recurrent Unit) structure and is capable of learning an active location variant structure for the recurrent connections.

Apart from doing the temporal modeling using recurrent units, a popular approach has been to use fully convolutional neural
networks, often two-dimensional, thus avoiding the modeling of explicit temporal dependencies. These networks have often
been based on U-Net-type architectures, one early example of which is the model by Agrawal et al. (2019), which predicts
the exceedance of rainfall over three distinct intensity thresholds for a one hour lead time. A more useful model is RainNet by
Ayzel et al. (2020). RainNet nowcasts rainfall continuously one timestep at a time, inserting the predicted frames back into the





network in order to make multiple lead time predictions. Similarly FureNET by Pan et al. (2021) nowcasts rainfall one hour at a time using polarimetric input variables in addition to observed rain rates, via multiple encoder branches and late fusion in the decoder of a residual U-Net architecture and brings improvement compared to using plain rain rates.

The principal problem of using discriminative deep learning models for deterministic precipitation nowcasting is that of the increasing blurring of nowcasts with increasing lead time. This is the natural consequence of attempting to minimize the pixel-wise forecasting error in the presence of uncertainties inherent to the task of predicting precipitation. Such loss functions thus behave in the same fashion as S-PROG and STEPS explicitly filtering out scales through their loss of predictability. One way to resolve the problem is to use generative modeling, which is the one taken by Ravuri et al. (2021) with their

Deep Generative Model of Radar (DGMR). DGMR is an adversarially trained convGRU-based generative model, capable of generating realistic time series of future radar observations, that outperform both classical and deep learning baseline models. In addition to deterministic nowcasts, DGMR is also capable of making ensemble-based probabilistic nowcasts.

Making probabilistic precipitation nowcasts using deep learning has been explored less than deterministic nowcasts, despite of the clear benefit of the probabilistic approach in operational use. In addition to DGMR, other existing probabilistic models

are MetNet (Sønderby et al., 2020) and its successor MetNet-2 (Espeholt et al., 2022). MetNet aggregates weather radar, satellite, and orographic information over a large area to predict a probability distribution of rain rate per pixel in one forward pass for a single lead time, with an architecture consisting of a spatial aggregator of inputs, a convLSTM spatial encoder, and a spatial decoder with axial attention. The model is shown to outperform the HRRR NWP model on an F1 metric for lead times up to 8 hours. MetNet-2 improves upon its predecessor by adding data assimilation context as an input and aggregating data

over a larger area. This enables it to outperform or at worst rival HRRR and HREF models in CRPS and CSI metrics for lead times up to 12 hours.

## 1.3   Uncertainty quantification and Bayesian deep learning

In addition to playing an important role in precipitation nowcasting, the importance of uncertainty quantification (UQ) has also been recognized in deep learning (Abdar et al., 2021). In the field of machine learning, the uncertainty of predictions

can be divided into two separate components: epistemic and aleatoric uncertainty. Epistemic uncertainty represents the lack of knowledge in the model, and it is reducible through improving the model or bringing in more training data. Aleatoric uncertainty on the other hand is inherent to the input data, and no amount of additional training data or model improvement will reduce it. Aleatoric uncertainty that varies over the input data is said to be heteroscedastic; a constant uncertainty is called homoscedastic.

Many approaches to the quantification of uncertainty have been developed on the deep learning side. One particularly important theme driving the development in this realm has been operational safety and countering overconfident predictions made by black box models overfitting the training data. Bayesian neural networks (BNN) have emerged as a candidate for addressing that issue. They work by placing probability distributions over the weights, which are estimated via the means of Bayesian inference and yield a predictive distribution for data through their marginalization.





Although exact Bayesian inference is intractable for large neural networks, suitable approximations exist. These are commonly divided into Markov Chain Monte Carlo (MCMC) and variational inference (VI) based methods (Jospin et al., 2022). MCMC methods predict better weight distributions but are more computationally expensive and thus often reserved for small-scale problems, where performance is key. VI on the other hand is more scalable and has been applied to larger neural networks. The idea behind variational inference is to approximate the true posterior of weights with a simpler analytic one (the variational

posterior), and to estimate the variational posterior which is the closest to the true one. Thanks to advances by Graves (2011) and subsequently Blundell et al. (2015) with the Bayes-By-Backprop (BBB) algorithm, it is now possible to use mini-batch optimization for mean-field VI (i.e., assuming fully factorizable variational posteriors) on large networks, opening up possibilities for the use of VI in problems such as precipitation nowcasting, that require large amounts of input data and numerous model parameters.

Later, Monte Carlo Dropout (Gal and Ghahramani, 2016) techniques among other variants have been identified as being equivalent to approximate Bayesian inference, losing some model expressivity but gaining ease of implementation. Based on this, Kendall and Gal (2017) have developed a technique for estimating the epistemic and heteroscedastic aleatoric variance components separately in deep learning regression tasks. They estimate the epistemic uncertainty with the variance of predictions made via Monte Carlo Dropout, and add a separate component to their network for predicting the aleatoric component.

The predictions are modeled as having Gaussian likelihoods, with means equal to the prediction point estimates and variances equal to the aleatoric term described. These terms are then learned by minimizing a Gaussian Negative Log likelihood loss function taking them and observations as inputs. This approach has recently started to be applied to problems such as the segmentation of satellite images (Dechesne et al., 2021), remaining useful life prognostics (Caceres et al., 2021), and long-term synoptic scale precipitation forecasts (Xu et al., 2022).

**1.4    Model idea and research questions**

We propose the **D**eep **E**nsemble-based **U**ncertainty **C**ombining radar **E**cho nowcasting (**DEUCE**) model for probabilistic precipitation nowcasting. The idea of the model is to apply the aleatoric and epistemic decomposition of uncertainty by Kendall and Gal (2017) to a Bayesian Convolutional neural network with mean-field variational inference for producing ensemble nowcasts of weather radar echo images, which represent the reflected power to the radar, often indicative of precipitation. The

research questions to which we will attempt to answer are the following:

1. *Can we produce both powerful and reliable ensemble precipitation nowcasts using Bayesian neural networks with uncertainty decomposition?* Specifically, is such a model competitive against classical baseline models when assessed with a variety of quantitative probabilistic prediction skill metrics as well as based on a qualitative assessment?

2. *What are the characteristics of the aleatoric/epistemic decomposition?* We are interested in the evolution of uncertain-

ties with prediction lead time, and whether they capture different and complementary features of the total predictive uncertainty.



3. *Can the model additionally be useful in producing deterministic precipitation nowcasts by means of averaging multiple predictions, leveraging the regulatory effect of probability distributions placed on weights?* Do such predictions perform competitively when assessed against classical baseline models using quantitative verification metrics? Also, what can those metrics tell about the nature of the predictions?

## 2   Model description

DEUCE builds upon a U-Net-based convolutional neural network (CNN) model of deterministic precipitation nowcasting, and turns it into a Bayesian neural network with variational inference for making the predictions stochastic, enabling us to model the uncertainty of this U-Net model. As mentioned, we build upon the work of Kendall and Gal (2017) for quantifying the uncertainty of the nowcasting task. Particularly, DEUCE attempts to decompose predictive uncertainty into aleatoric uncertainty (originating from data, irreducible) and epistemic uncertainty (induced by lacking knowledge, reducible) by predicting reflectivity fields along with the aleatoric uncertainty associated with them explicitly. Epistemic uncertainty in turn is estimated from the variance of the reflectivity fields sampled, and it is combined with aleatoric uncertainty at inference time in order to yield an approximation of the total predictive uncertainty.

### 2.1   Functional model

A neural network $f_{\boldsymbol{\theta}}(\boldsymbol{x}) = \hat{\boldsymbol{y}}$ is a universal function approximator, which can be used for regression tasks, mapping an input quantity $\boldsymbol{x}$ to an output value $\hat{\boldsymbol{y}}$ approximating the real value $\boldsymbol{y}$ using its parameters $\boldsymbol{\theta}$. In the case of radar-based precipitation nowcasting with neural networks, we approximate a function mapping the spatio-temporal time series of past radar observation images $\boldsymbol{x} = \boldsymbol{x}_1, \boldsymbol{x}_2, \ldots, \boldsymbol{x}_{L_{\text{in}}}$, where $L_{\text{in}}$ corresponds to the number of input timesteps, to future radar observation images $\boldsymbol{y} = \boldsymbol{y}_1, \boldsymbol{y}_2, \ldots, \boldsymbol{y}_{L_{\text{out}}}$, where $L_{\text{out}}$ corresponds to the number of input timesteps. In the DEUCE model, both $\boldsymbol{x}$ and $\boldsymbol{y}$ represent processed radar reflectivity data, and the network $f_{\boldsymbol{\theta}}(\boldsymbol{x}) = \hat{\boldsymbol{y}}, \boldsymbol{\sigma}^2$ outputs a tuple of predicted reflectivity field time series $\hat{\boldsymbol{y}} = \hat{\boldsymbol{y}}_1, \hat{\boldsymbol{y}}_2, \ldots, \hat{\boldsymbol{y}}_{L_{\text{out}}}$ along with fields estimating the aleatoric uncertainties $\boldsymbol{\sigma}^2 = \boldsymbol{\sigma}_1^2, \boldsymbol{\sigma}_2^2, \ldots, \boldsymbol{\sigma}_{L_{\text{out}}}^2$ corresponding to each of the pixels of $\hat{\boldsymbol{y}}$.

For the task of precipitation nowcasting, the neural network has to be capable of outputting predictions for multiple *lead times*, i.e., discrete time steps in the future, corresponding to future radar observations. DEUCE achieves this by using a variant of U-Net as its functional architecture, taking in a sequence of 12 radar reflectivity fields $\boldsymbol{x}$, predicting $\hat{\boldsymbol{y}}, \boldsymbol{\sigma}^2$ corresponding to the nowcast for the next 12 timesteps in a single forward pass.

A schematic representation of the the main components of DEUCE and how they are connected is presented in Fig. 1. The architecture consists of a single encoder branch, extracting features from $\boldsymbol{x}$ at different spatial scales and semantic levels. The feature maps from these different scales are preserved for later use through skip-connections. The (largest scale) latent state produced by the encoder, as well as intermediate feature maps mediated by skip-connections, are then fed to two independent decoders: one outputting $\hat{\boldsymbol{y}}$ and the other outputting $\log \boldsymbol{\sigma}_{al}^2$. Using separate decoders for the outputs is preferable over a single combined decoder to avoid the blending of adjacent features, which would be detrimental to the expressivity of the model.





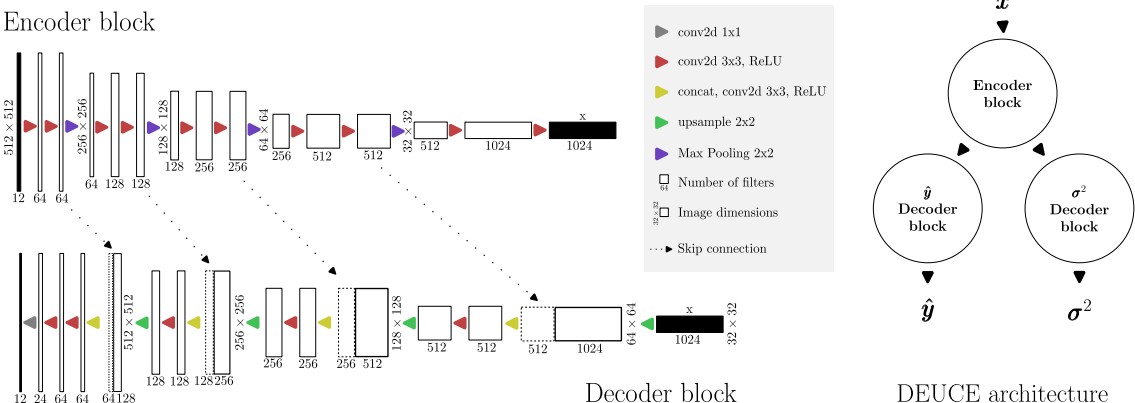

**Figure 1.** The DEUCE encoder and decoder architectural components depicted on the left, along with the architectural diagram making use of those components on the right. Feature maps at different scales are extracted in the encoder branch, before being passed to decoder branches, providing the outputs of the network.

The network contains two-dimensional (spatial) convolutions. These are represented by `conv2d 3x3` and `conv2d 1x1`
labels denoting layers with filter sizes 3 and 1, respectively. By using 2D convolutions, temporal dependencies are only present implicitly. This approximation casts the nowcasting task as a simple image sequence-to-sequence translation problem, which reduces the computational resources needed compared to explicit modeling of the temporal aspect. The convolutional layers use *partial convolutions* (Liu et al., 2018), in which missing values are masked and only valid values are used to normalize the convolutions. Although we do not work with missing data, this design choice helps in providing better quality predictions
near image borders by reducing, e.g., various artifacts related to them. `ReLU` denotes the activation function of the same name, `concat` concatenation along the channel dimension, `upsample` nearest-neighbor upsampling by a scale of 2, and `Max Pooling` maximum pooling by a scale of 2.

## 2.2 Stochastic model

Conventional neural networks are deterministic in their nature, meaning that they only ever yield the same output $\hat{y}$ for a given
input $x$ and parameters $\theta$. Our goal is to produce a reliable estimate of the uncertainty associated with the approximation produced by the neural network. Because this approximation is merely a function of the input data and the functional model including parameters, considering the uncertainty of these sources separately should allow the approximation of the total predictive uncertainty of nowcasts.

Hence, epistemic uncertainty is modeled by placing probability distributions on functional model parameters $\theta$, effectively
turning the model stochastic. A Bayesian approach is taken in this regard, placing a prior distribution upon the weights, and estimating the most likely posterior distribution given that prior and the training data. The estimation of the true posterior is an intractable task for a large-scale neural network, which is why variational inference (VI) is used to learn approximate





posterior estimates for weights. VI limits the space of acceptable posterior distributions to a parameterized family, whose learned parameters replace the point estimates of classical neural network weights. Here, we aim to minimize the Kullback-Leibler (KL) divergence (Kullback and Leibler, 1951) between the true and variational posteriors, which is a measure of the similarity between two probability distributions. As such, the objective is stated as

$$\boldsymbol{\theta}^* = \arg\min_{\boldsymbol{\theta}} D_{\mathrm{KL}}[q(\boldsymbol{w} \mid \boldsymbol{\theta}) \| p(\boldsymbol{w} \mid \mathcal{D})] \tag{1}$$

where $\boldsymbol{\theta}$ denotes the variational posterior parameters, $\boldsymbol{\theta}^*$ the optimal parameters, $\boldsymbol{w}$ the sampled network weights, $\mathcal{D} = (\boldsymbol{x}, \boldsymbol{y})$ the problem data, $q(\boldsymbol{w} \mid \boldsymbol{\theta})$ the variational posterior, and $p(\boldsymbol{w} \mid \mathcal{D})$ the exact posterior of network weights. In practice, this is not directly solvable, so the optimization is accomplished through the maximization of an evidence lower bound (ELBO) proxy objective. The objective is defined as

$$\mathrm{ELBO}(\mathcal{D}, \boldsymbol{\theta}) = \mathbf{E}_{q(\boldsymbol{w}|\boldsymbol{\theta})}[\log p(\boldsymbol{w}, \mathcal{D})] - \mathbf{E}_{q(\boldsymbol{w}|\boldsymbol{\theta})}[\log q(\boldsymbol{w} \mid \boldsymbol{\theta})] \tag{2}$$

$$= \overbrace{\mathbf{E}_{q(\boldsymbol{w}|\boldsymbol{\theta})}[\log p(\mathcal{D} \mid \boldsymbol{w})]}^{\text{likelihood}} + \underbrace{\overbrace{\mathbf{E}_{q(\boldsymbol{w}|\boldsymbol{\theta})}[\log p(\boldsymbol{w})]}^{\text{prior}} - \overbrace{\mathbf{E}_{q(\boldsymbol{w}|\boldsymbol{\theta})}[\log q(\boldsymbol{w} \mid \boldsymbol{\theta})]}^{\text{posterior}}}_{=-\mathbf{E}_{q(\boldsymbol{w}|\boldsymbol{\theta})} D_{\mathrm{KL}}[q(\boldsymbol{w}|\boldsymbol{\theta}) \| p(\boldsymbol{w})], \text{i.e., the complexity term}}, \tag{3}$$

consisting of the log-likelihood, log-prior, and log-posteriors, with the last two terms commonly grouped together as the complexity term. Here $\mathbf{E}_{q(\boldsymbol{w}|\boldsymbol{\theta})}$ denotes the expected value of the probability density of interest over the variational posteriors.

According to Blundell et al. (2015), in Bayesian neural networks and using mini-batch optimization, the ELBO objective as stated in Eq. 3 can be approximated as

$$\mathrm{ELBO}_i^{\pi}(\mathcal{D}_i, \boldsymbol{\theta}) \approx \frac{1}{N} \sum_{n=1}^{N} \Big( \log p(\mathcal{D}_i \mid \boldsymbol{w}_{i,n}) + \pi_i \log p(\boldsymbol{w}_{i,n}) - \pi_i \log q(\boldsymbol{w}_{i,n} \mid \boldsymbol{\theta}) \Big), \tag{4}$$

which acts as an unbiased Monte Carlo estimator of the ELBO, and is our final loss function. Here, the cost is calculated for each $i$:th of the $M$ mini-batches in an epoch, each time drawing $N$ Monte Carlo samples of the variational posteriors of weights. $\pi_i$ denotes an arbitrary weighting of the complexity term, using in this work the same rule as in Blundell et al. (2015), which is $\pi_i = 2^{M-i}/(2^M - 1)$. This serves to make the regularization effect of the prior stronger earlier, allowing data to be more important later in the training. In DEUCE, the variational posterior distributions $q$ are modeled as diagonal Gaussian distributions, and the Bayes-By-Backprop (BBB) algorithm using the re-parametrization trick by Blundell et al. (2015) is employed for their optimization. The prior distribution $p(\boldsymbol{w})$ on the contrary is fixed as a hyperparameter, and is identically as well as independently distributed for each parameter as a normal distribution with zero-mean and a variance of 0.1. This allows us to potentially calculate the complexity cost in closed form (Hershey and Olsen, 2007), rather than with the Monte Carlo estimate of Eq. 4, hence reducing the computational cost of training.

The likelihood cost of Eq. 4, similarly to Kendall and Gal (2017), is modeled for the $i$:th mini-batch and the $n$:th Monte Carlo sample as the Gaussian log likelihood

$$\log P(\mathcal{D} \mid \boldsymbol{w}) = -\frac{1}{P} \sum_{p=1}^{P} \frac{1}{2} e^{-\boldsymbol{s}_p} (\boldsymbol{y}_p - \hat{\boldsymbol{y}}_p)^2 + \frac{1}{2} \boldsymbol{s}_p, \tag{5}$$





where the cost is averaged over $p = 1 \ldots P$ pixels of the $L_{out} \times W \times H$ spatio-temporal time series $\boldsymbol{s}$, $\boldsymbol{y}$ and $\hat{\boldsymbol{y}}$; $L_{out}$ refering to the length of the time series, $W$ to the width of the images, and $H$ to the height of the images. $i$ and $n$ indices of fields are omitted here for clarity. Here, $\boldsymbol{y}$ denotes the observed reflectivity fields, $\hat{\boldsymbol{y}}$ the predicted reflectivity fields using the $n$:th

weights sampled from the network, and $\boldsymbol{s} := \log \boldsymbol{\sigma}^2$ refers to the corresponding logarithm of the aleatoric variances predicted by the network with those weights. The logarithm of the aleatoric variances estimate is taken because optimizing using it is more computationally stable, and was found to work better than simply using variance constrained to be positive with a ReLU output activation function, especially dealing with variances approaching zero.

## 2.3 Generation of ensemble nowcasts

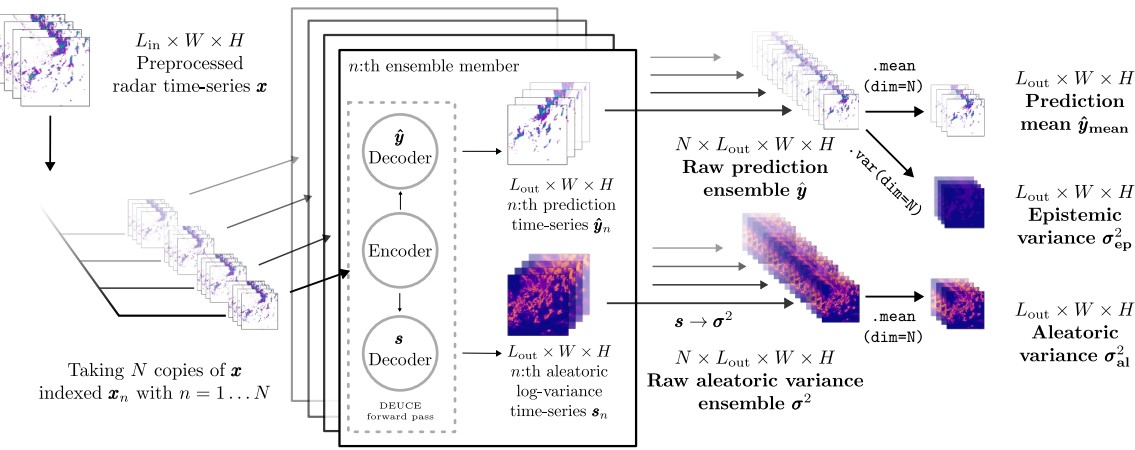

**Figure 2.** The prediction procedure for the primary outputs of the DEUCE model illustrated. Each sampled output is computed separately with a forward pass through the network, yielding a time series of the predictions and the logarithmic aleatoric variances, which are converted back to variances. After agglomeration into a pair of raw 'ensembles', the prediction mean $\hat{\boldsymbol{y}}_{\mathrm{mean}}$, as well as the two types of uncertainties, the epistemic variances $\boldsymbol{\sigma}_{\mathrm{ep}}^2$ and aleatoric variances $\boldsymbol{\sigma}_{\mathrm{al}}^2$ are computed from the pair. These three quantities are the ones used for producing the final prediction ensemble.

The procedure for producing the primary outputs of DEUCE for making probabilistic nowcasts is presented in Fig. 2. First, $N$ raw network outputs are produced, which are stochastic pairs of reflectivity field sequences $\hat{\boldsymbol{y}}_n$ and logarithmic aleatoric variance field sequences $\boldsymbol{\sigma}_n$. Each $n$:th of those sampled outputs draws different weights from the learned variational posterior distributions, which is reflected in the output distribution. The $\boldsymbol{\sigma}_n$ are converted to their non-logarithmic version $\boldsymbol{\sigma}_n^2$, and individual stochastic runs are stacked into a pair of raw 'ensembles' $\hat{\boldsymbol{y}}, \boldsymbol{\sigma}^2$. At this point, the epistemic uncertainty is embedded

in $\hat{\boldsymbol{y}}$, but the aleatoric uncertainty is separate, only present in $\boldsymbol{\sigma}_{\mathrm{al}}^2$. Hence, in order to allow for the combination of these uncertainties, $\hat{\boldsymbol{y}}$ is divided into the prediction mean $\hat{\boldsymbol{y}}_{\mathrm{mean}}$ and the epistemic variance $\boldsymbol{\sigma}_{\mathrm{ep}}^2$ by taking the mean and variance over $\hat{\boldsymbol{y}}$ respectively. Additionally, the aleatoric variance $\boldsymbol{\sigma}^2$ is summarized by taking its mean, denoted $\boldsymbol{\sigma}_{\mathrm{al}}^2$. These three outputs: $\hat{\boldsymbol{y}}_{\mathrm{mean}}$, $\boldsymbol{\sigma}_{\mathrm{ep}}^2$, and $\boldsymbol{\sigma}_{\mathrm{al}}^2$ form the base from which probabilistic nowcasts are computed.





The total mean and uncertainty of the prediction can thus be estimated as

$$\hat{\boldsymbol{y}}_{\mathrm{mean}} = \frac{1}{N}\sum_{n=1}^{N}\hat{\boldsymbol{y}}_n, \qquad \boldsymbol{\sigma}_{\mathrm{pred}}^2 \approx \overbrace{\frac{1}{N}\sum_{n=1}^{N}\hat{\boldsymbol{y}}_n^2 - (\frac{1}{N}\sum_{n=1}^{N}\hat{\boldsymbol{y}}_n)^2}^{\boldsymbol{\sigma}_{\mathrm{ep}}^2} + \overbrace{\frac{1}{N}\sum_{n=1}^{N}\boldsymbol{\sigma}_n^2}^{\boldsymbol{\sigma}_{\mathrm{al}}^2}, \qquad (6)$$

where $\boldsymbol{\sigma}_{\mathrm{pred}}^2$ denotes the predictive variance. This means that the predictive variance can be estimated as the sum of the variance of the predicted reflectivity fields, which is the epistemic variance, and of the mean of the predicted aleatoric variance fields. These quantities are sufficient for making probabilistic nowcasts such as calculating exceedance probabilities for precipitation intensity, as we model the predictive distribution of precipitation as normally distributed with mean $\hat{\boldsymbol{y}}_{\mathrm{mean}}$ and variance $\boldsymbol{\sigma}_{\mathrm{pred}}^2$.

Nevertheless, some applications of probabilistic precipitation nowcasting — such as flood modeling — assume ensemble-based nowcasts where each member of the ensemble represents a physically plausible precipitation scenario. One could of course randomly sample the predictive distribution to generate an ensemble, which would correctly approximate pixel-wise statistics, but the spatio-temporal structure of the fields would be lost. In an attempt to remedy to this, we post-process outputs to generate ensemble members respecting the spatial covariance structure of the input field $\boldsymbol{x}$ as

$$\hat{\boldsymbol{y}}_n^{\mathrm{ens}} = \hat{\boldsymbol{y}}_{\mathrm{mean}} + \sqrt{\boldsymbol{\sigma}_{\mathrm{pred}}^2} \otimes \boldsymbol{\epsilon}_{\mathrm{corr},n}, \qquad (7)$$

where $\hat{\boldsymbol{y}}_n^{\mathrm{ens}} = \hat{\boldsymbol{y}}_{n,1}^{\mathrm{ens}}, \hat{\boldsymbol{y}}_{n,2}^{\mathrm{ens}}, \ldots, \hat{\boldsymbol{y}}_{n,L_{\mathrm{out}}}^{\mathrm{ens}}$ denotes the newly generated ensemble member, $\otimes$ an element-wise multiplication broadcast over $L_{\mathrm{out}}$ frames, and $\boldsymbol{\epsilon}_{\mathrm{corr},n}$ is a correlated Gaussian random field of shape $W \times H$. $\boldsymbol{\epsilon}_{\mathrm{corr},n}$ is generated to match the average spatial correlation structure of $\boldsymbol{x}$ using Fast Fourier Transform (FFT) filtering. The structure is obtained non-parametrically from the power spectrum of $\boldsymbol{x}$ (Seed et al., 2013). The technique is equivalent to that used to generate perturbation fields in

STEPS (Pulkkinen et al., 2019). Even though this method accounts for the spatial structure of the precipitation time series, it is not capable of modeling its temporal structure, which is assumed constant. The ensembles produced this way shall be denoted $\hat{\boldsymbol{y}}^{\mathrm{ens}}$, in contrast to the raw predicted reflectivity fields denoted $\hat{\boldsymbol{y}}$.

## 3 Experimental details

This section presents the experiments performed. First, in Sect. 3.1, we present the dataset used, followed by the details related
to the training of DEUCE in Sect. 3.2, and the verification experiments in Sect. 3.3. Additional technical details can on the other hand be found in Sect. A.

### 3.1 Data

The dataset used for this work comes from the Finnish Meteorological Institute radar network. It consists of cropped lowest-altitude radar reflectivity composites, chosen from rainy days during the summer period of years 2019–2021. The dataset is
identical to that used by Ritvanen et al. (2023), only using longer time series. The composites are built from the two lowest





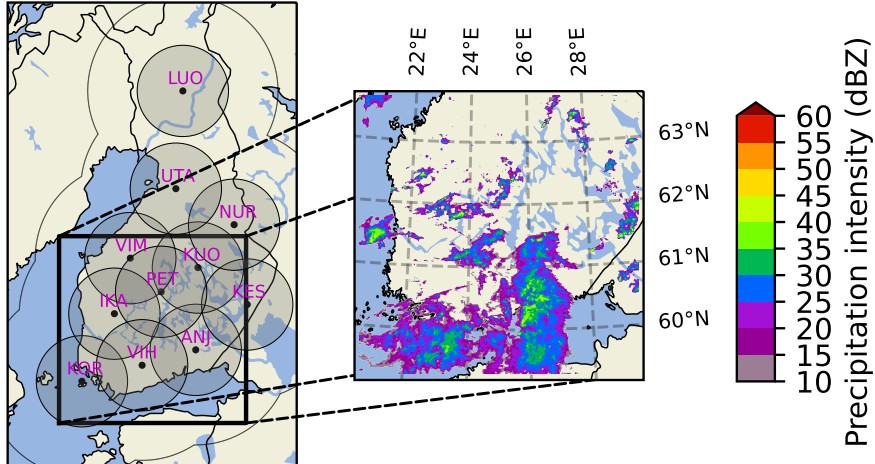

**Figure 3.** The Finnish Meteorological Institute radar network with its 11 radars and the bounding box used. Each radar is described by its three-letter code, with their 120km coverage radii for snowfall in gray and the intersection of 250km coverage radii for rainfall as the black outline. An example radar composite crop from a precipitation event (15 August 2019 at 15:00:00 UTC) is visualized in the zoom onto the bounding box in the right.

elevation angle scans, interpolated into an 1x1 km Cartesian grid. The chosen area covers southern Finland, with the bottom left corner at coordinates (59.01°N, 20.55°E) and the top right corner at coordinates (63.62°N, 30.27°E). The spatial extent of this crop is 512x512 km, corresponding to 512x512 pixel square images, suitable for training a neural network. The composites are available with a temporal resolution of five minutes. The extent of the bounding box is additionally illustrated in Figure 3

along with the coverage of Finnish Meteorological Institute radars. From this, we see that the advantage of the crop is that it has a higher density radar cover than its surroundings.

The data was selected on a day-by-day basis, selecting the 100 days with the most pixels having reflectivity values over 35 dBZ. The days were then divided into six hour long blocks, from which blocks with less than one percent of pixels with reflectivity values over 20 dBZ were removed. These remaining blocks were then randomly split into training, validation, and

verification datasets with a ratio of 6:1:1. The division into blocks was done in order to limit the number of successive time series present in different splits, as they exhibit high correlation, and not using any blocks would make the training, validation, and verification sets dependent as the same events would be present in all of them. Six hours was deemed a sufficient time for temporal correlations to mostly disappear. Lastly, two-hour long time series, corresponding to 24 images each, were then extracted from these blocks using a sliding window principle, with a stride of one, omitting those time series with missing data.

The final training, validation, and verification datasets ended up containing 10780, 1813, and 1666 time series respectively.

The input time series were read from HDF5 files, stored there with an 8-bit scale-offset lossy compression scheme, ranging from -32 to 96 dBZ at a resolution of 0.5 dBZ. The images were then converted to floating point values and a threshold of 8 dBZ was applied, replacing values below the threshold with -10 dBZ. This served as a simple way to remove non-meteorological



targets and other clutter that could interfere with the training and prediction, while maintaining most of the relevant precipitation
echoes. Finally, the reflectivity values were normalized between zero and one. Computed predictions were converted back into
reflectivity values by applying the inverse of the transformation, before saving them using the same scheme as with the input
data.

## 3.2 Training

For the training of the network, the Adam optimizer (Kingma and Ba, 2015) was used with an initial learning rate of 1e-4 and
other parameters set to their PyTorch default values. The network was trained with that learning rate for 20 epochs, after which
the learning rate was lowered to 1e-5 for 8 more epochs, and finally further lowered to 1e-6 for one final epoch. A validation
epoch was carried out after each epoch, in which Equivalent Threat Score (ETS) (Hogan et al., 2010) metrics were calculated
for converted precipitation estimates (Sect. A1) of predictions, and summed over thresholds of 0.5, 1.0, 5.0, 10.0, 20.0, and
30.0 mmh$^{-1}$ as well as each lead time. This validation score showed improvement over the whole training process.

The training procedure for DEUCE is presented in Fig. 4 for a single epoch. Both input sequence lengths $L_{\text{in}}$ as well as
output sequence lengths $L_{\text{out}}$ were 12, corresponding to one hour each. Both for the training and validation epochs, the batch
size was set to two and the number of produced Monte Carlo samples of posteriors $N$ to two as well, which was the most that
our GPU could fit during training. In order to increase the variance between the gradients of mini-batch members, Flipout re-
parametrization (Wen et al., 2018) was applied to the sampled weights, multiplying the random sampling coefficient of weights
with a random sign matrix, effectively adding randomness inside batches for a low computational cost. The closed-form of the
KL divergence between two Gaussian distributions was used for the calculations of the ELBO complexity term instead of
Monte Carlo estimates in the final model training.

The input time series $X_i$ was pre-processed first as described in Section 3.1, and in the case of training data, was then
augmented by applying in succession a random horizontal flip, a random vertical flip, and a rotation by an angle randomly
chosen between 0, 90, 180, and 270 degrees. This was done to improve the variety of the training dataset and consequently
improve the generalization performance of the trained network.

## 3.3 Verification

The performance of the DEUCE model is verified against the pySTEPS (Pulkkinen et al., 2019) implementation of multiple
extrapolation-based precipitation methods. The verification is divided into the qualitative inspection of ensembles produced in
a case study, into an analysis of DEUCE uncertainty composition, into the verification of the (probabilistic) performance of the
whole ensemble, and the verification of the (deterministic) performance of the ensemble mean, i.e., its fidelity in representing
the true variation of the radar images. The four types of verification performed, along with the relevant DEUCE product, the
baseline models used, and the evaluation criteria are summarized in Table 1.

In probabilistic verification experiments, $N = 48$ ensemble members are used both for producing the raw outputs $\hat{\boldsymbol{y}}, \boldsymbol{\sigma}^2$
and for drawing the post-processed ensemble $\hat{\boldsymbol{y}}^{\text{ens}}$, as well as for making the baseline ensemble model predictions. All of the
predictions made for the verification of DEUCE are made until 60 minute lead time and thresholded at 8 dBZ, serving as an



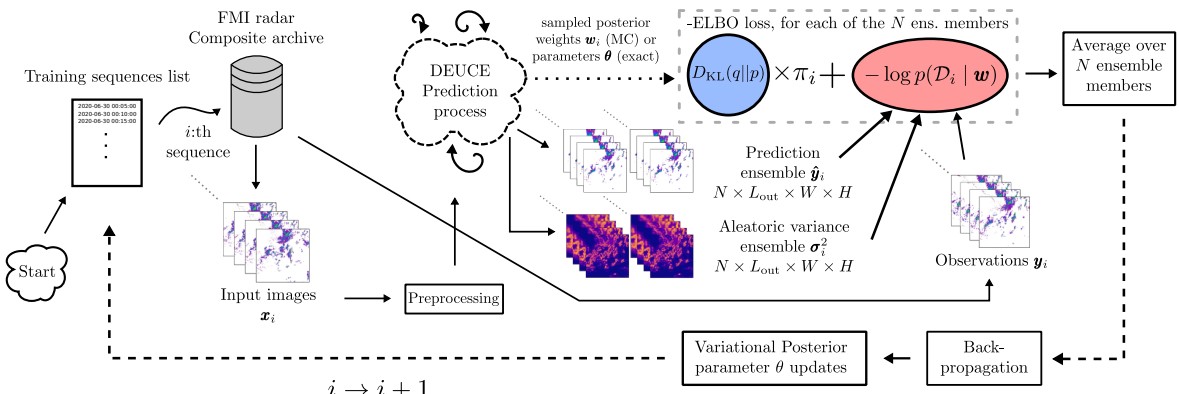

**Figure 4.** A training epoch for DEUCE illustrated. One loop corresponds to a single training sequence, which can be substituted for a single training mini-batch, taking multiple sequences in one batch. The DEUCE prediction process refers to that illustrated in Fig. 2, without the post-processing. The blue box labeled $D_{\mathrm{KL}}(q\|p)$ corresponds to the complexity term of the negative ELBO loss (to minimize), $\pi_i$ to its weighting coefficient, and the red box labeled $-\log p(\mathcal{D}_i \mid \boldsymbol{w})$ to the likelihood term of the negative ELBO loss. Monte Carlo estimates of the complexity term use sampled weights $\boldsymbol{w}_i$, whereas the closed-form expression that we use is a function of parameters $\boldsymbol{\theta}$.

| | *DEUCE product* | *Baseline models (Sect. A2)* | *Evaluation criteria* |
|---|---|---|---|
| Case study (Sect. 3.3.1) | $\hat{\boldsymbol{y}}^{\mathrm{ens}}$ | STEPS, LINDA-P | Ensemble mean/STD, exceedance probabilities |
| Uncertainty composition (Sect. 3.3.2) | $\hat{\boldsymbol{y}},\boldsymbol{\sigma}^2$ | - | case decomposed mean/STD & statistics |
| Probabilistic perf. (Sect. 3.3.3) | $\hat{\boldsymbol{y}}^{\mathrm{ens}}$ | STEPS, LINDA-P | CRPS, Reliability diagram, ROC AUC, Rank hist. |
| Deterministic perf. (Sect. 3.3.4) | $\hat{\boldsymbol{y}}_{\mathrm{mean}}$ | Extrapolation, LINDA-D | ME, ETS, RAPSD |

**Table 1.** The four components of the verification process for DEUCE summarized.

estimate for minimum observable precipitation. Four precipitation thresholds are considered where the verification involves evaluating the quality of a prediction exceeding a particular reflectivity value. Converted using the Z-R relationship presented in Sect. A1, these are 20 dBZ ($\approx 0.5 \ \mathrm{mmh}^{-1}$), 25 dBZ ($\approx 1.3 \ \mathrm{mmh}^{-1}$), 35 dBZ ($\approx 5.7 \ \mathrm{mmh}^{-1}$), and 45 dBZ ($\approx 25.5 \ \mathrm{mmh}^{-1}$), which correspond to very light, light, moderate, and heavy rain respectively.

### 3.3.1 Case study

A challenging rainfall event is chosen as a case study to provide a qualitative assessment, as well as a comparison, of DEUCE nowcasts against the baseline probabilistic methods. The case study focuses on an ensemble nowcast at a particular timestep during the precipitation event, chosen such as to include both large-scale weaker precipitation, characteristic of stratiform rainfall, and localized heavy precipitation, characteristic of convective rainfall. The latter has a shorter lifetime and has been





traditionally harder to predict, but it is of interest to observe the performance of the model with both types. In addition, we choose a case allowing us to observe instances of both weakening and intensification of echoes. The case is chosen from radar composite crops with the area described in Sect. 3.1 over the summer of the year 2022, which is separate from the dataset used for training, validation, and quantitative verification. The timestamp of the chosen case is the 9 July 2022 at 15:00:00 UTC.

The radar images of the hour leading at 15:00 UTC are used as inputs and the following hour is predicted.

Three different visualizations of the case are made at 5, 15, 30, and 60 minute lead times, using the post-processed DEUCE ensembles $\hat{\boldsymbol{y}}^{\mathrm{ens}}$, and the probabilistic baseline models STEPS and LINDA-P described in Sect. A2 when appropriate. The first visualization is that of predictive means and standard deviations of the ensembles in dBZ units. Here, DEUCE, STEPS, and LINDA-P are compared side-by-side. The second visualization is that of exceedance probabilities of DEUCE, STEPS, and

LINDA-P ensemble nowcasts at a 25 dBZ reflectivity threshold. The third and last of the visualizations depicts the exceedance probability of DEUCE in predicting reflectivity above 20, 25, 35, and 45 dBZ thresholds.

### 3.3.2 Uncertainty composition analysis

The composition of the DEUCE predictive uncertainty is analyzed both using the prediction of the case study, and using statistics aggregated over the verification dataset. For the case study prediction, the aleatoric and epistemic components of

the predictive standard deviation are visualized next to the combined predictive uncertainty, the mean predictions, and the observations, at lead times of 5, 15, 30, and 60 minutes. The statistics collected are average magnitude of the aleatoric and epistemic standard deviation components under different conditions. These magnitudes are divided into bins corresponding to the prediction lead time and the observed reflectivity matching the pixel in question (5 dBZ bin width from 5 to 60 dBZ), and are collected for each prediction timestamp. The resulting statistics are visualized in the form of a histogram aggregated over

the whole dataset, and as bar plots showing the contribution of the uncertainties against lead time and observed reflectivity.

### 3.3.3 Probabilistic performance verification

Probabilistic verification serves to assess the probabilistic predictive power of DEUCE ensembles, mostly in terms of prediction reliability and discrimination ability. In other words, it determines the quality and the variety of produced ensembles with regard to the true distribution of different future scenarios. Here, the DEUCE prediction is represented by the post-processed ensemble

$\hat{\boldsymbol{y}}^{\mathrm{ens}}$. Probabilistic baseline models used are STEPS (Bowler et al., 2006; Seed et al., 2013) and LINDA-P (Pulkkinen et al., 2021). The description and configuration of those models are given in Sect. A2.

The probabilistic performance metrics used are the Continuous Ranked Probability score (CRPS) (Hersbach, 2000; Wilks, 2011), which generalizes the Mean Absolute Error of deterministic forecasts to probability distributions, and is calculated for lead times until 60 minutes. Next, the Receiver Operating Characteristic (ROC) curve (Mason, 1982; Wilks, 2011) along

with the area under it (AUC) quantify the discriminative power of the ensembles for predicting reflectivity values exceeding a certain threshold. ROC AUC is computed for reflectivity thresholds of 20, 25, 35, and 45 dBZ at lead times of 5, 15, 30, and 60 minutes. For measuring forecast reliability and sharpness, we used the reliability diagram along with its sharpness histogram (Wilks, 2011) as well as the Expected Calibration Error (ECE) score (Naeini et al., 2015), which we all computed for the



same threshold and lead times as ROC curves. Finally, rank histograms (Wilks, 2011) were calculated for measuring the bias
and spread of ensembles at lead times of 5, 15, 30, and 60 minutes. A detailed description of these metrics along with the
configurations used is found in Sect. A3.

### 3.3.4 Deterministic performance verification

Deterministic verification serves to assess whether DEUCE ensemble means are useful themselves. It also gives insight into
many interesting aspects of predictions, such as systematic biases and the possible loss of small-scale variability. Here, the
DEUCE prediction is represented by the ensemble mean $\hat{y}_{\mathrm{mean}}$. Deterministic baselines used are an extrapolation nowcast and
LINDA-D (Pulkkinen et al., 2021). The description and configuration of those models is again described in Sect. A2.

Three deterministic metrics are used to assess DEUCE ensemble means. The first is the Mean Error (ME) (Wilks, 2011),
measuring the bias of nowcasts produced. The Equitable Threat Score (ETS) (Hogan et al., 2010; Wilks, 2011) then provides an
estimate of the deterministic skill in forecasting precipitation above a certain intensity threshold. It is calculated for lead times
up to 60 minutes and thresholds of 20, 25, 35, and 45 dBZ. Finally, the Radially-Averaged Power Spectral Density (RAPSD)
(Ruzanski and Chandrasekar, 2011; Ulichney, 1988) measures how well the power spectrum of precipitation is maintained. It is
summarized with a relative MAE score. We compute RAPSD for prediction lead times of 5, 15, 30, and 60 minutes. RAPSD is
also calculated for individual $\hat{y}^{\mathrm{ens}}$ members to analyse the possible contribution of the spatially correlated noise to maintaining
the power spectrum. A detailed description of these metrics along with the configurations used is found in Sect. A4.

## 4  Results

The results of the quantitative and qualitative analyses of model performance and fitness to the task indicate that DEUCE
succeeds in its primary task of providing reasonably reliable probabilistic precipitation nowcasts, but not in that of producing
skillful deterministic nowcasts. This is illustrated by the summary of quantitative verification results is provided in Table
2. These results will then be elaborated in detail and presented as figures in the following four subsections. Starting by the
qualitative case study results in Sect. 4.1, we then present the composition of the uncertainty in Sect. 4.2, before continuing
with the probabilistic performance metric results in Sect. 4.3, and finally presenting the deterministic performance metric
results in Sect. 4.4.

### 4.1  Case Study

The results of the case study in Fig. 5, 6, and 7 suggest that DEUCE ensemble nowcasts are able to give reasonable uncertainty
and exceedence probability estimates at multiple thresholds and lead times, and that DEUCE nowcasts look similar to those
given by STEPS, albeit being less grainy. Figure B1 shows an example of what individual ensemble members look like at
different prediction lead times. The predictions all start quite similar, but they eventually diverge, driven by the increasing
predictive uncertainty and different patterns of correlated noise. The ensemble members exhibit variety while preserving a



| | Probabilistic models | | | | Deterministic models | | |
|---|---|---|---|---|---|---|---|
| | DEUCE (ours) | STEPS | LINDA-P | | DEUCE mean (ours) | Extrapolation | LINDA-D |
| CRPS ↓ | 1.29 | **1.27** | 1.43 | AME ↓ | 1.31 (-) | **0.35** (-) | 0.53 (+) |
| ECE 20 ($\times 10^3$) ↓ | **6.88** | 9.45 | 13.36 | ETS 20 ↑ | 0.442 | 0.435 | **0.454** |
| ECE 25 ($\times 10^3$) ↓ | **5.36** | 6.64 | 8.44 | ETS 25 ↑ | 0.299 | 0.341 | **0.371** |
| ECE 35 ($\times 10^3$) ↓ | 1.97 | **1.13** | 2.04 | ETS 35 ↑ | 0.047 | 0.134 | **0.162** |
| ECE 45 ($\times 10^4$) ↓ | - | - | - | ETS 45 ↑ | 0.006 | 0.049 | **0.056** |
| ROC AUC 20 ↑ | **0.968** | 0.957 | 0.943 | RAPSD rel. MAE 5 ↓ | 0.55 | **0.08** | 0.39 |
| ROC AUC 25 ↑ | **0.960** | 0.938 | 0.926 | RAPSD rel. MAE 15 ↓ | 0.74 | **0.08** | 0.52 |
| ROC AUC 35 ↑ | **0.885** | 0.784 | 0.840 | RAPSD rel. MAE 30 ↓ | 0.84 | **0.07** | 0.58 |
| ROC AUC 45 ↑ | **0.706** | 0.610 | 0.689 | RAPSD rel. MAE 60 ↓ | 0.90 | **0.11** | 0.65 |

**Table 2.** Quantitative verification metrics summarized. ↑ indicates that a higher score is better, while ↓ indicates that a lower score is better. The best score amongst models is marked using a **bold** font. ECE scores indicate the Expected Calibration Error, an aggregate measure of reliability. AME stands for Absolute Mean Error and RAPSD rel. MAE score summary values indicate the relative Mean Absolute Error between the PSD of observation and predictions. For AME, the sign of the mean error is reported in parentheses. Scores are averaged over lead times for which they were calculated, except for RAPSD rel. MAE scores, in which they are averaged over frequencies. Numerical values in ECE, ROC AUC, and ETS score names indicate dBZ threshold values, and in RAPSD lead time in minutes. ECE 45 results are omitted because results are not comparable for cause of missing data in some of the bins of the DEUCE reliability diagrams.

moderate amount of realism, nevertheless limited by the increasing smoothing of the predictive mean and variance fields with lead time.

### 4.1.1 Ensemble mean and breadth

Ensemble mean and breadth as units of standard deviation is shown in Figure 5. Here, we can see in all models a trend towards loss of predicted reflectivity intensity and disappearance of heavily localized echoes. However, these are in all models compensated by an increase in the spatial extent and the magnitude of the ensemble standard deviation. In LINDA-P, the effect of predicting in rain rate units ($\mathrm{mmh^{-1}}$) is seen as the uncertainty of cell borders emphasized. LINDA-P also generally exhibits smaller and more uniform standard deviation than the other models. For a one hour lead time, DEUCE seems to have a generally an ensemble breadth a bit smaller than STEPS but higher than LINDA-P, with the most heterogeneity in standard deviation values.

### 4.1.2 Reflectivity exceedance probabilities

Reflectivity probabilities of exceeding 25 dBZ for the case predicted by the different models are shown in Fig. 6. Overall, DEUCE seems to provide balanced exceedance probabilities, not missing any significant areas even after one hour, but not covering excessively large areas. Comparatively, STEPS tends to completely miss some significant portions such as in the area



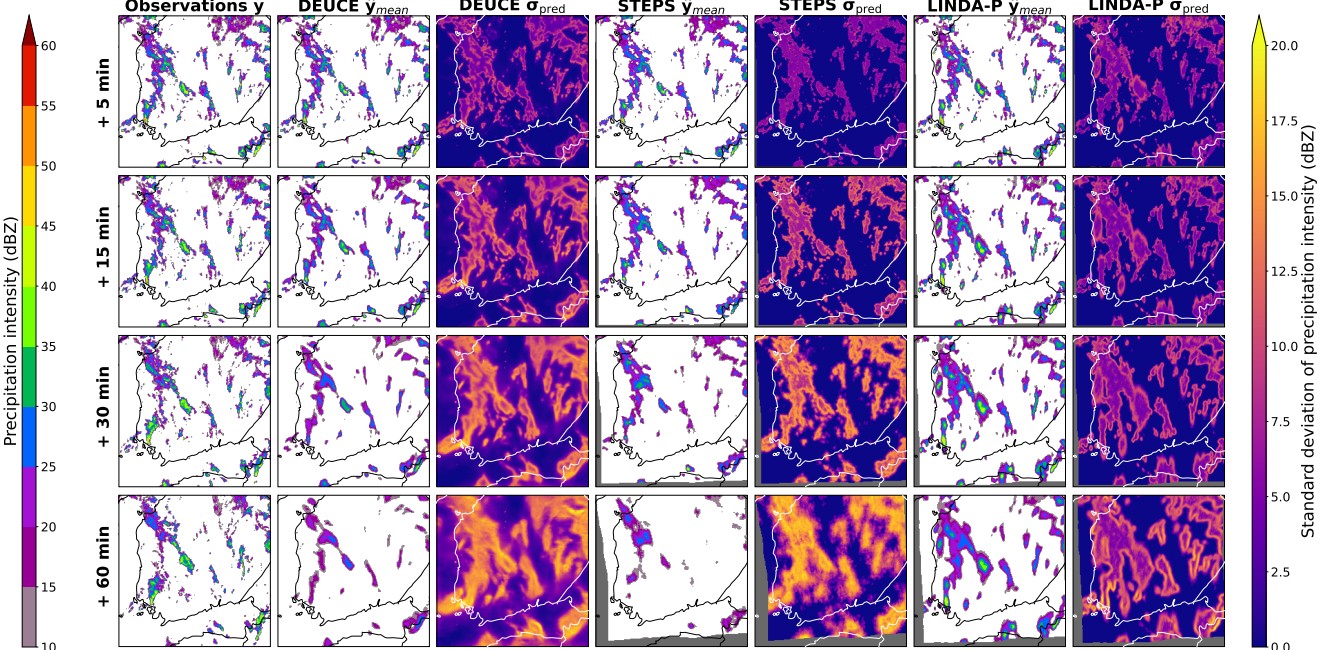

**Figure 5.** Ensemble means and breadths of DEUCE compared against STEPS and LINDA-P model predictions and observations for multiple lead times. The area covers southern Finland, starting at 15:00:00 UTC on the 9 July 2022. The rows represent lead time and columns different instances of observations, model mean and standard deviations. Missing values are indicated by a dark gray color.

highlighted in the south-west of Finland at one hour, and generally seems to predict eventually smaller probabilities for the evolution of smaller cells. LINDA-P on the other hand suffers from overconfidence and misplaces the evolution of multiple

precipitation areas after one hour. On a general level for all models compared, the advection field is well captured, while the growth and decay of echoes is often not very effectively forecast. The anisotropic structure of the uncertainty shown through exceedance probabilities is also much better captured by DEUCE and LINDA-P than STEPS. In addition, because it is not based on the extrapolation of radar echoes, there are no "dead zones" filled with NaN values (dark gray color) and DEUCE is able to provide nowcasts to varying success in border regions where STEPS and LINDA-P predictions are not necessarily

defined.

Lastly, The exceedance probabilities of DEUCE nowcasts for 15, 25, 35, and 45 dBZ reflectivity thresholds for the present case are shown in Fig. 7. We can see that DEUCE is able to nowcast an exceedance probability at all thresholds (which are indeed all exceeded at some place and point in the observations). Higher thresholds exhibit lower values and some misplacement of exceedance probabilities, as precipitation exceeding those are more difficult to predict and have smaller areas.



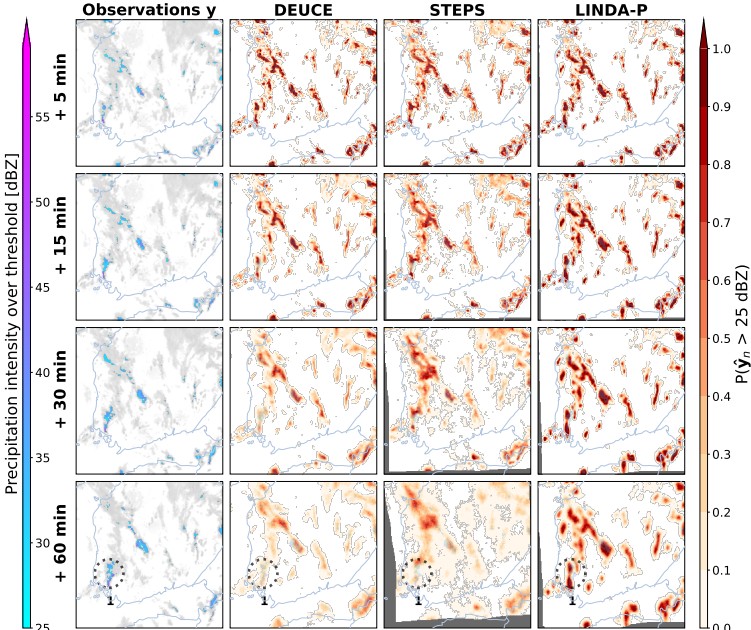

**Figure 6.** Reflectivity exceedance probabilities of 25 dBZ for DEUCE against STEPS and LINDA-P model predictions and observations for multiple lead times. The area covers southern Finland, starting at 15:00:00 UTC on 9 July 2022. The rows represent lead time. In the leftmost column, actual threshold-exceeding precipitation is shown in colors, with non-exceeding precipitation additionally shown faintly in light gray in the background. In other columns, threshold exceeding precipitation is again shown overlayed with exceedance probabilities of models in shades of red. Missing values are indicated by a dark gray color. The circles labeled 1 highlight a case of DEUCE model improvement over baselines.

## 4.2 Analysis of the aleatoric and epistemic uncertainty dichotomy

The relative contribution of aleatoric and epistemic uncertainty for the case outlined previously is presented in Fig. 8. We can see that most of the predictive uncertainty in fact comes from the aleatoric part. Epistemic uncertainty is of much smaller magnitude, and its contribution is further reduced when working in terms of variance in the calculation of predictive uncertainty. We can see that epistemic uncertainty does not extend as much away from the core of predicted reflectivity as aleatoric uncertainty, which reflects a small variance in the raw $\hat{y}$ ensemble.

A more detailed view into the contribution of aleatoric and epistemic components is provided in Fig. 9, with statistics over the whole verification dataset. A histogram of the uncertainties aggregated over all lead times and observed reflectivity values is shown on the left of Fig. 9. Epistemic uncertainty has a very narrow distribution mostly between 0–5 dBZ, which means that its average value could not have varied much in different cases, lead times, and observed reflectivity values, pointing to small model uncertainty response to these factors. Aleatoric uncertainty on the other hand has a long-tail distribution, centered



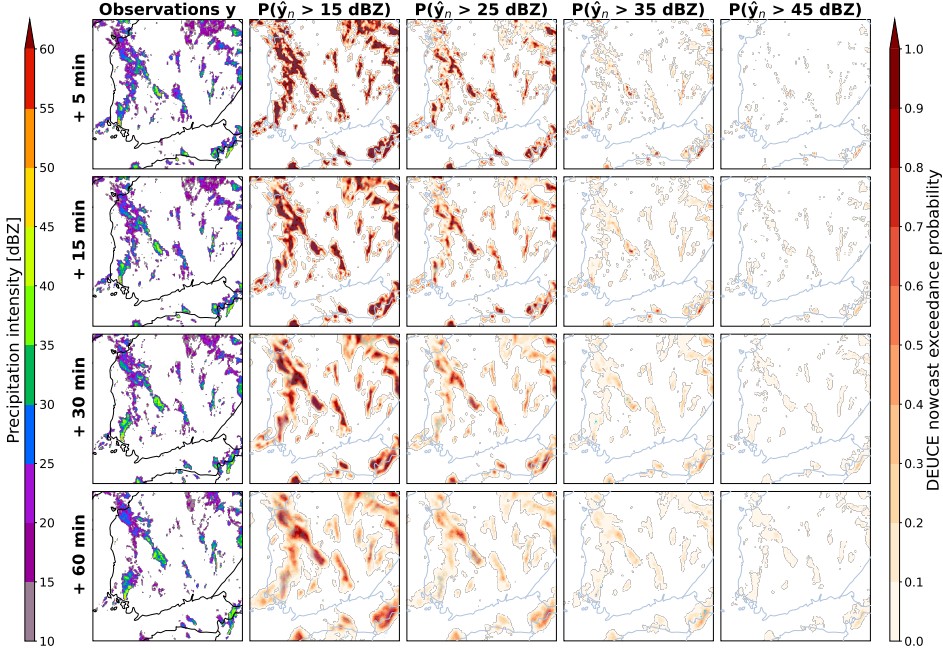

**Figure 7.** Reflectivity exceedance probabilities of DEUCE against observations for multiple lead times and precipitation intensity thresholds. The area covers southern Finland, starting at 15:00:00 UTC on 9 July 2022. The rows represent lead time. The leftmost column is observations, and the rest are exceedance probabilities at different thresholds. As in Fig. 6, observations exceeding the threshold in question are plotted in shades of gray, overlayed with probabilities in shades of red.

around 10 dBZ, but going up until values over 30 dBZ, which keeps open the possibility for a dependence on these external factors.

The suspicions are confirmed when inspecting the bar plots on the right of of Fig. 9, where mean aleatoric uncertainty shows a clear dependence on prediction lead time and to some degree on observed reflectivity. Aleatoric uncertainty seems to clearly
increase with lead time and to first slightly decrease, before increasing again in relation to observed reflectivity. One possible explanation to this last observation is that reflectivity values below 20 dBZ often correspond to the edges of precipitation cells, which are difficult to predict, and that reflectivity values over 35 dBZ often correspond to heavy precipitation with short lifetime and thus bad predictability. In between those, there is more predictable precipitation patterns, such as the interior of stratiform precipitation cells. Epistemic uncertainty on the other hand does not seem to show any particular dependence on prediction lead
time, which might have to do with the fact that the model predicts all lead times at once, making it possibly more difficult for the predictions to vary depending on lead time. There is on the other hand some very slight increase of epistemic uncertainty with observed reflectivity, which might be an accurate reflection of the relatively smaller amount of training data available for high observed reflectivity values.





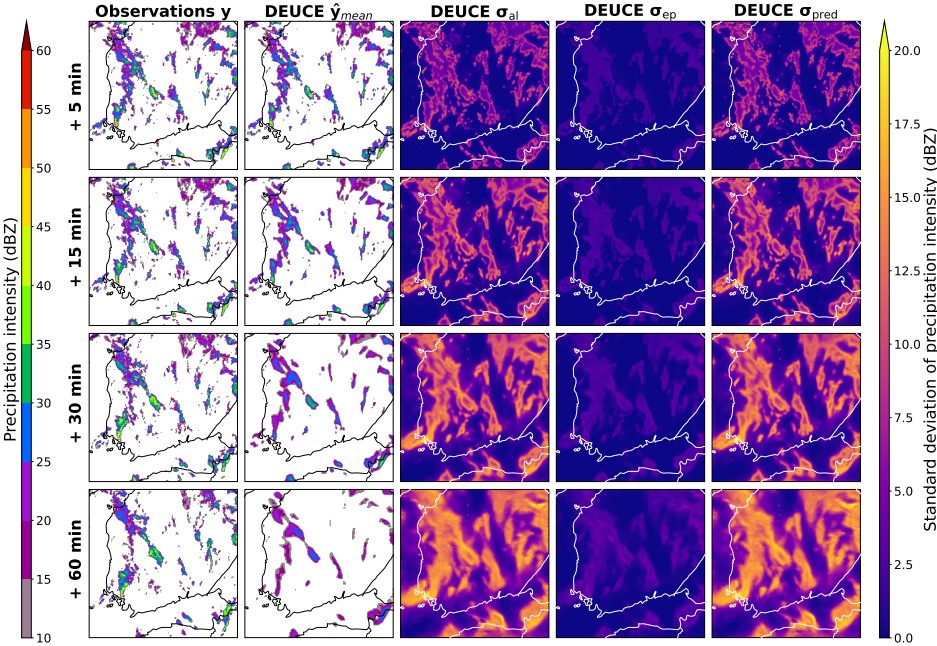

**Figure 8.** Composition of the predictive uncertainty for the case study. The area covers southern Finland, starting at 15:00:00 UTC on 9 July 2022. The rows represent lead time and columns observations (ground truth), the mean prediction, aleatoric and epistemic standard deviation components, and the combined predictive standard deviation.

### 4.3 Probabilistic skill verification

The reliability diagrams and sharpness histograms for probabilistic nowcasts are depicted in Fig. 10. It can first be noted that in general, DEUCE nowcasts are very close to the dashed black line indicating a perfectly reliable forecast. Sometimes, this is to a similar degree as baseline models, but in some cases, such as a long lead time and a high threshold, DEUCE is closer to the diagonal than baselines. This is however not reflected in the ECE scores at 35 dBZ, shown in Table 2, as smaller forecast probabilities are weighted much higher due to their sample count here, making STEPS the most reliable model at 35 dBZ

by this metric. An important pattern is that compared to baseline models, DEUCE is prone to slight under-forecasting of the exceedance probabilities. This is particularly the case for a short lead time (5 min), where the effect is the most pronounced. As lead times grow longer and thresholds get higher, nowcasting gets harder and there is an overall tendency in all models, but particularly LINDA-P, to over-forecast threshold exceedance.

From the sharpness histogram, it is seen that the distribution of forecast probabilities is more or less uniform at low thresh-
olds, but more biased towards small exceedance probabilities at higher thresholds. These higher thresholds are where the difference between DEUCE and baselines are visible, there being a considerably lower number of cases of high forecast probability in DEUCE than in baselines.





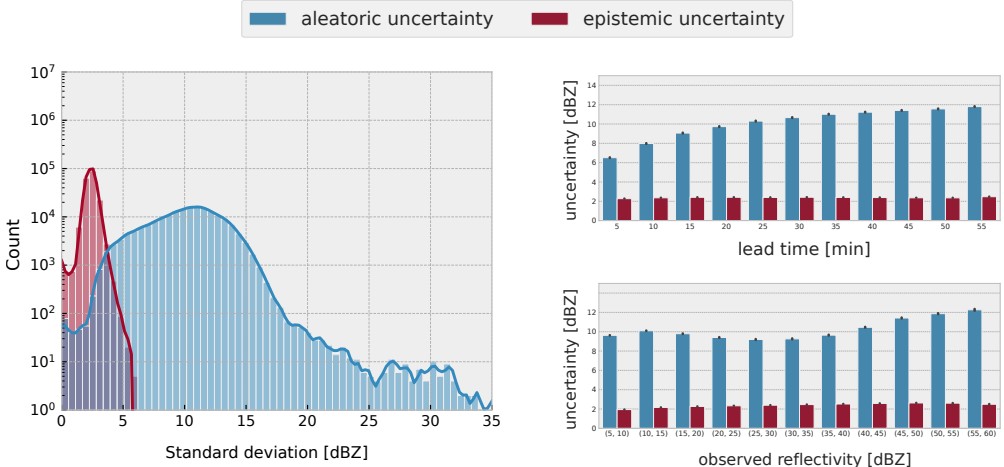

**Figure 9.** Visualization of the statistics on the composition of predictive uncertainty over the verification dataset. On the left, a histogram of aleatoric and epistemic standard deviation (SD) aggregated over all lead times and observed reflectivity values is shown. On the right, we arrange the same data into bar plots to show the relationship of the type of the uncertainty SD with prediction lead time (top) and observed ground truth radar reflectivity (bottom).

The rank histogram of nowcasts is shown in Figure 11. It is apparent here that DEUCE is constantly slightly biased towards predicting too low reflectivity values, and that the spread is large at short lead times, but less significant later on. STEPS exhibits a very balanced flat histogram, but LINDA on the other hand has a U-shaped histogram, characteristic of a too small ensemble breadth in general.

The results for the ROC area under the curve probabilistic nowcast metric are shown in Fig. 12. In this benchmark DEUCE achieves the best results at all thresholds. We can notice that STEPS has good discriminative power at low thresholds but that it does not scale well to higher ones, and that LINDA-P is not competitive at lower thresholds but excels as the threshold grows. Nevertheless, DEUCE manages to perform better than both in their skillful areas.

Lastly, the CRPS verification metric is depicted in Fig. 13. It can be seen that the lowest and best score is achieved by STEPS at all lead times. DEUCE comes then second, slightly above STEPS, and LINDA-P lags far behind. Overall with CRPS, it can be seen that DEUCE achieves adequate results in the order of baseline models.

From the quantitative probabilistic verification, it can be summarized that DEUCE achieves satisfactory and well-rounded performance. The model does not significantly lack in any category in particular, and offers a good trade-off between forecast reliability and discriminatory power.





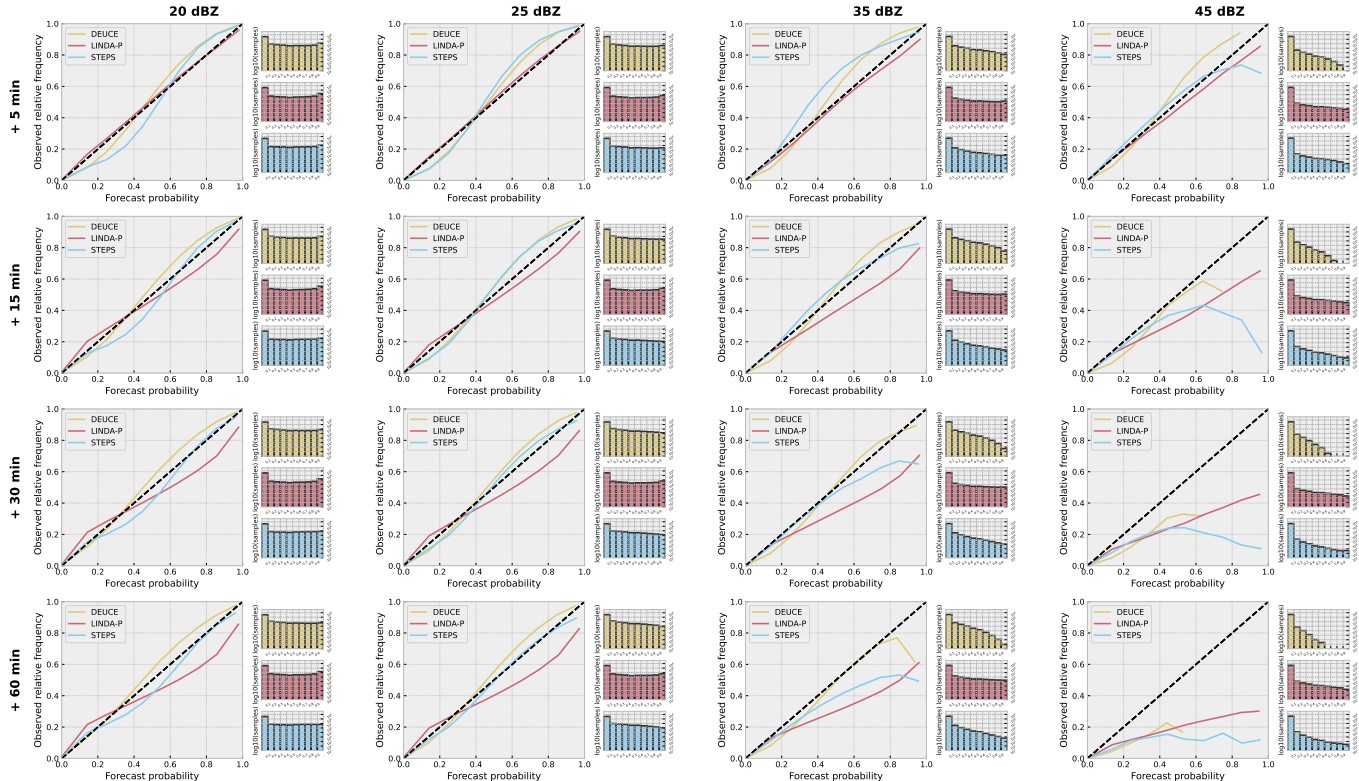

**Figure 10.** The reliability diagrams and sharpness histograms for DEUCE (yellow), STEPS (blue), and LINDA-P (red) model nowcasts at exceedance probability thresholds of 20, 25, 35, and 45 dBZ at lead times of 5, 15, 30, and 60 minutes. Rows indicate lead time and columns the exceedance probability threshold. The diagonal dashed black lines indicate perfect reliability.

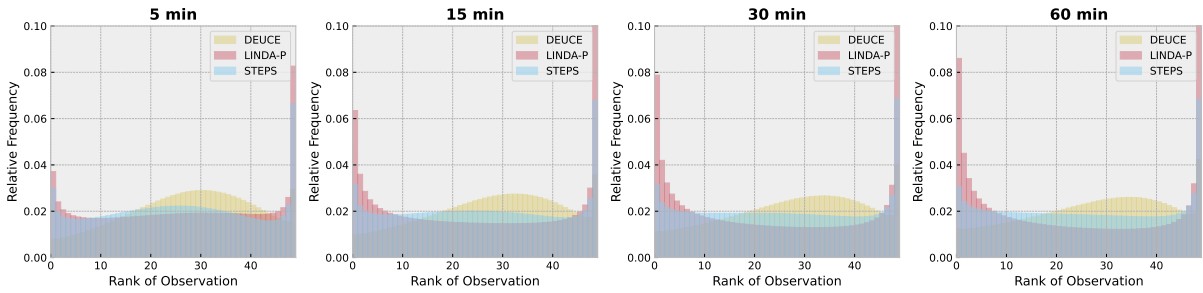

**Figure 11.** Rank histograms of ensemble nowcasts, including DEUCE, STEPS, and LINDA-P, at lead times of 5, 15, 30, and 60 minutes over the verification set.




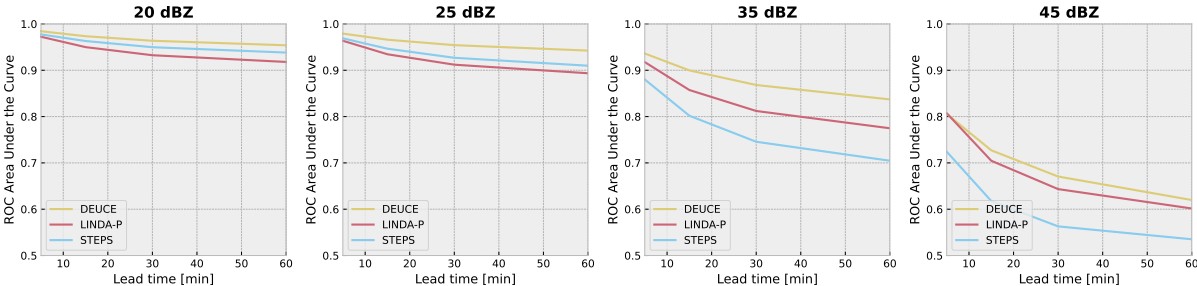

**Figure 12.** The ROC Area under the Curve (AUC) values at lead times until 60 minutes for DEUCE (yellow), STEPS (blue), and LINDA-P (red) model nowcasts at exceedance probability thresholds of 20, 25, 35, and 45 dBZ.

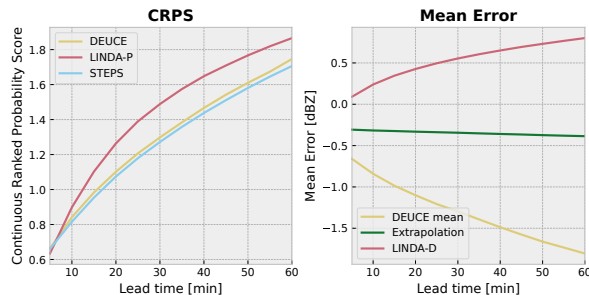

**Figure 13.** The CRPS score of DEUCE (yellow) against the ensemble baselines STEPS (blue) and LINDA-P (red) is shown on the left. The Mean Error (ME) score for non-augmented ensemble mean predictions of DEUCE (yellow) against deterministic baseline extrapolation (green) and LINDA-D (red) nowcasts is shown on the right.

## 4.4 Deterministic skill verification

Here, we analyse the results of the comparison of the deterministic nowcast skill between DEUCE non-augmented mean predictions $\hat{y}_{\mathrm{mean}}$ and baseline predictions. First off, a depiction of the mean nowcasting error (ME) until a 60 minute lead time is presented in Fig. 13. While extrapolation nowcasts have on average a ME slightly below zero, DEUCE is more strongly negatively biased, while LINDA-D is strongly positively biased.

Further, the equivalent threat score (ETS) results for precipitation thresholds of 20, 25, and 35 dBZ are shown in Fig. 14. The ETS score of DEUCE is competitive for the 20 dBZ threshold, but at 25 dBZ its progression at lead times longer than 30 minutes is already worse than baselines. At 35 and 45 dBZ, DEUCE already performs worse than baselines at any lead time examined. The reason for this weakness is the compound effect of intrinsic CNN prediction smoothing and the averaging of ensemble members. This smoothing effect is also visible in the Radially-averaged power spectral density (RAPSD) results for nowcasts, presented in Fig. 15. Average RAPSD is computed for nowcasts at lead times of 5, 15, 30, and 60 minutes. It can clearly be seen that compared to baselines, the fields predicted by DEUCE lose more power at small spatial scales, and that this effect is heavily amplified at longer lead times, which again illustrates the above compound effect. However, this effect



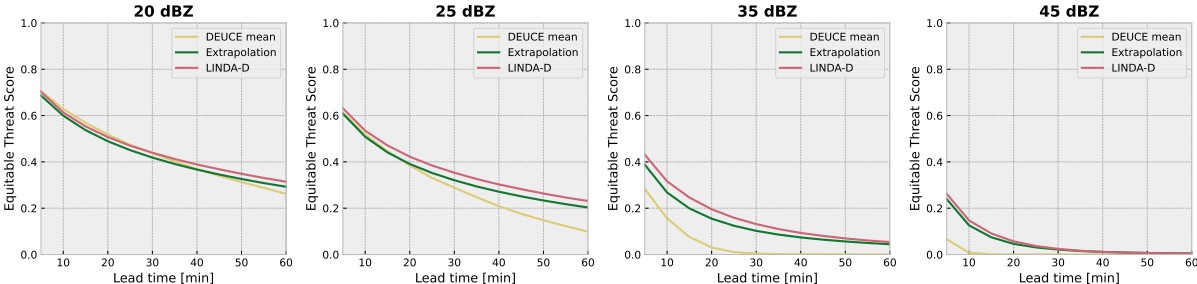

**Figure 14.** Equivalent Threat Scores (ETS) as a function of lead time for non-augmented DEUCE ensemble means (yellow) compared against those of extrapolation (green) and LINDA-D (red) deterministic baseline models for reflectivity thresholds of 20, 25, 35, and 45 dBZ at lead times until 60 minutes. DEUCE ensemble means perform competitively for predicting precipitation over 20 dBZ, but see their relative performance drop at higher reflectivity thresholds.

475    seems to be damped by augmenting the mean prediction with uncertainty-weighted correlated noise following the structure of the input field, especially at longer lead times.

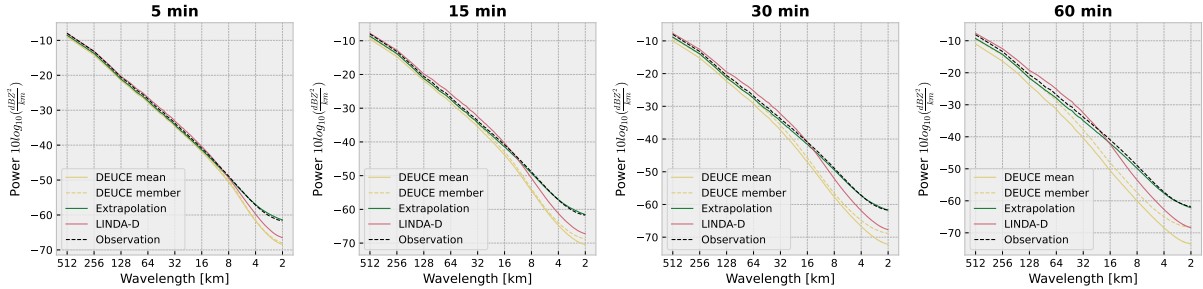

**Figure 15.** Radially averaged power spectral density (RAPSD) for non-augmented DEUCE ensemble means (yellow solid), individual augmented DEUCE ensemble members (yellow dashed) compared against those of extrapolation (green) and LINDA-D (red) deterministic baseline models at lead times of 5, 15, 30, and 60 minutes. The observation RAPSD is shown as a black dashed line.

## 5   Discussion

DEUCE probabilistic ensemble nowcasts proved to be both relatively reliable and skillful compared to STEPS and LINDA. Nevertheless when analysing the reliability diagram, it is apparent that DEUCE is under-confident especially at short lead

480    times. Further inspection of rank histograms (Fig. 11)— showing the distribution of the rank of observations among ensemble members— indicated that this under-confidence is expressed by 1) a too large ensemble spread, and 2) a slight bias towards ensembles producing too weak estimates, as was also visible in the ensemble mean Mean Error score. The too large ensemble spread may be a byproduct of attempting to predict the ensemble mean and aleatoric variances all at once, with the prior placed



on weights placing a limit on the complexity of the model, in effect privileging the learning of longer lead times where the
average errors are of bigger magnitude.

We can also observe that most of the uncertainty is of aleatoric nature, and that the contribution of epistemic variance is universally low. In addition to having trained with a large amount of data, low epistemic variance is probably related to the variational inference mechanism, as the combination of Bayes-By-Backprop and Flipout reparametrization have been shown to yield too small epistemic uncertainty estimates by Valdenegro-Toro and Mori (2022), when compared to MC Dropout, Deep Ensembles, and Markov Chain Monte Carlo. Another factor that might have played a role in this is again predicting all lead times at once, because adopting the iterative approach of RainNet (Ayzel et al., 2020) would have propagated previous epistemic uncertainties to subsequent lead times, possibly balancing out the contributions, also allowing making predictions for an arbitrary number of timesteps. On the other hand, abandoning the recursive prediction scheme of RainNet significantly reduces the time complexity of computing $\hat{\boldsymbol{y}}, \boldsymbol{\sigma}^2$ from $\mathcal{O}(NL)$ to $\mathcal{O}(N)$ where $N$ is the sample size and $L$ is the number of prediction lead times.

Contrary to the preliminary iteration of the model focusing only on modeling epistemic uncertainty (Harnist, 2022), the current model better captures the increased spread of the predictive distribution with lead time through the aleatoric component. An alternative model — without a separate decoder branch for $\boldsymbol{\sigma}^2$ and only allocating a separate output channel to it — was not successful because the $\hat{\boldsymbol{y}}$ and $\boldsymbol{\sigma}^2$ that it learned were highly correlated, more blurry, and lacked expressivity. It is for this reason that the two-branch version was adopted.

In DEUCE, small-scale variability is steadily lost with increasing lead time. This is a problem for the production of realistic nowcasts, as loss of small-scale variability is synonymous to loss of information. However, it can be justified in the case of a probabilistic model, assuming that the variety of the ensemble is preserved despite of the loss of high frequencies. In DEUCE, the predictive distribution is modeled explicitly, so this loss is not a major problem for us. In the preliminary version of the DEUCE model (Harnist, 2022) however, this loss had adverse effects as the model was trained with homoscedastic (fixed as a hyperparameter) aleatoric uncertainty modeling which was not taken into account when making predictions, resulting in a too small ensemble spread. Hence, despite the retention of small-scale variation not being completely necessary, better conservation of those components may help retain expressivity in a deep learning nowcasting model, as has been seen through the success of generative modeling, showcased by the DGMR results of Ravuri et al. (2021). Although limited, the spatially correlated noise scheme for sampling the predictive uncertainty may help integrate DEUCE into applications where physically plausible ensemble members are necessary. Still, the lack of temporal correlation modeling inside of post-processed ensemble members and the smoothing of the predictive means and variances themselves limit realism, pointing to the limits of the taken approach.

With regard to the model development in general, some amount of hyper-parameter optimization was performed. Those hyper-parameters related to the functional model as well as the optimizer are mainly inherited from RainNet (Ayzel et al., 2020), and those related to variational inference mostly originate from the preliminary version of the model (Harnist, 2022). There, the VI related parameters specifically demanded non-trivial tuning for model converge and acceptable result production, which might limit the immediate applicability of the model in its default state. Also, the local optimality of the current hyper-




parameters is not assured. Despite of this, VI and epistemic uncertainty are not decisive factors in the model performance, and swapping out those components is a potential way forward. Moreover, 60 minutes (12 frames) of input data and the same length for predictions was picked without optimization in an attempt to preserve some symmetry between the network inputs and outputs. Although there is yet no consensus on how many frames are needed, as little as four input frames have be enough to saturate model performance in some conditions (Ravuri et al., 2021), so tuning the ratio of input to output frames could be a viable thing to try.

Regarding the verification process, it is a pertinent question to ask whether the used baseline models were sufficient to validate the performance of DEUCE. Particularly, the lack of deep learning ensemble baselines is one weakness of the performed verification. It would have been particularly interesting to use the Deep Generative Model of Radar (DGMR) by Ravuri et al. (2021) as a baseline, as it represents the current state-of-the-art in deep learning based precipitation nowcasting, and is capable of producing ensemble nowcasts. Unfortunately, we were not able to successfully train DGMR on our dataset using the resources that we had allocated for the task. Other models of interest that were not included in the verification are MetNet by Sønderby et al. (2020) and its successor MetNet-2 by Espeholt et al. (2022), which use, e.g., orographic and satellite data in addition to radar data. We hope that further work will make possible the comparison of DEUCE probabilistic nowcasting performance to other deep learning based models.

One last point of concern is in the validity of the verification metrics used. The potential issues here mostly relate to the summarizing quantitative metrics of Table 2. Firstly, the relative RAPSD MAE metric for measuring power-spectrum fidelity of predictions uses a tighter sampling of points towards wavelengths representing small spatial scales, which biases it to give a higher weight to those scales. Although we are indeed mostly interested in small-scale variations, this property means that even big discrepancies in the power of large spatial scales will be under-represented. Next, the ECE metric used to summarize the reliability of ensemble models is very sensitive to variations in the order of magnitude of the number of samples per bin. This behaviour is significant especially at higher exceedance thresholds, where almost all prediction probabilities are concentrated in the smallest probability bin, giving almost no weight to even mildly successful nowcasting of rare but significant events of high heavy precipitation probability. This means that ECE doesn't necessarily provide a complete assessment of model reliability in the context of probabilistic precipitation nowcasting.

# 6 Conclusions

We developed a probabilistic precipitation nowcasting model named DEUCE, based on a Bayesian neural network with variational inference, and featuring the combination of epistemic and aleatoric uncertainty estimates in an attempt to yield reliable yet powerful probabilistic predictions. The model succeeded at this primary task, performing competitively against the baseline STEPS and LINDA-P models, judged both using qualitative and quantitative evaluation.

It was found that DEUCE had issues with the representation of epistemic uncertainty, leading to most of the uncertainty ending up appearing as aleatoric uncertainty, maybe due to the variational inference used. The aleatoric uncertainty exhibited a clear dependence on lead time and corresponding observed reflectivity, which are factors heavily influencing the predictability.



The epistemic uncertainty on the other hand showed little dependence on these factors, with the exception of a slight increase with observed reflectivity, which might reflect the distribution of the training data. Based on this, aleatoric and epistemic uncertainties do indeed seem to capture complementary features of the predictive uncertainty. Finally, the ensemble means were found to perform worse compared to extrapolation and LINDA-D baselines, showing that the model in its current state is not useful in the deterministic case due to the excessive smoothing of predictions.

Looking into future research directions, DEUCE has a number of different facets upon which its performance could be improved. First, the underlying U-Net could potentially be replaced by an more powerful architecture capable of modeling explicit temporal dependencies. The spatio-temporal extent could be enlarged, and additional orographic, polarimetric, or satellite input channels could improve parts of the nowcasts. It is possible to additionally try to leverage other patterns for increasing predictability, such as operating in Lagrangian coordinates as shown by Ritvanen et al. (2023), for increasing prediction performance. From a probabilistic aspect, certain alternative inference methods, such as Radial Bayesian Neural Networks (Farquhar et al., 2020) or Deep Ensembles look promising as a potential way to ease the training and improve the representation of epistemic uncertainty.

Regardless of its shortcomings, DEUCE is a first step in ensemble-based probabilistic precipitation nowcasting using Bayesian neural networks. The concurrent modeling of aleatoric and epistemic uncertainties has the potential to be useful for operational forecasters, and the model in its current state forms a strong yet relatively lightweight baseline for future developments in deep learning based probabilistic precipitation nowcasting.

*Code and data availability.* The data used for the production of the results is available online (Harnist et al., 2023) at https://doi.org/10. 23728/fmi-b2share.3efcfc9080fe4871bd756c45373e7c11. This data includes the input data used for the training of DEUCE, prediction generation, and observations for the verification. Pre-trained model checkpoints, the script used to gather neural network inputs into an HDF5 file, as well as computed metric data are also included.

The source code with instructions for the reproduction of results is available online (Harnist, 2023) at https://doi.org/10.5281/zenodo. 7961955 and on Github at https://github.com/fmidev/deuce-nowcasting. This code is used for the training and nowcast generation of DEUCE, the production of baseline nowcasts, the computation of metrics, and the creation of figures presenting these metrics.

## Appendix A: Additional technical details

### A1 Ground precipitation estimates from reflectivity

The formula $R = (10^{z/10}/223)^{1/1.53}$ was used in cases where an estimate of ground precipitation corresponding to lowest level radar reflectivity composites was needed. Here R denotes precipitation estimates in $mmh^{-1}$ units, and z denotes radar reflectivity in dBZ units. The parameters of the Z–R relationship employed in the formula come from the work of Leinonen et al. (2012) and aim to estimate the amount of rainfall corresponding to radar reflectivity measurements from Finnish Meteorological Institute polarimetric C-band radars in Finland.



## A2 Baseline models

There are two deterministic baseline models: a simple extrapolation nowcast and the deterministic variant of LINDA (LINDA-
D). The extrapolation nowcast extrapolates the last input reflectivity field along a motion field calculated from the last four
elements of the input time series. In the extrapolation nowcast and all other baseline methods, we use the dense Lucas-Kanade
optical flow method with its default pySTEPS parameters for the computation of the motion field. In addition, all baseline
nowcasting methods use the semi-Lagrangian integration scheme from pySTEPS for performing the extrapolation, with cubic
interpolation and other parameters left to their default values.

LINDA, being a more advanced extrapolation-based method capable of predicting high-intensity rainfall more accurately,
serves as a natural benchmark both in the deterministic and probabilistic cases, for the ability of the model to capture convective
rainfall evolution. LINDA predictions are made using reflectivity fields converted to rain rate using the method described in
Sect. A1, as it is required for the model to work. The LINDA models here use the last three input rain rate fields as input in
addition to the motion field. They do not use feature detection in order to reduce the prediction computation time over the
verification set to more practical durations. The ensemble-producing version of LINDA: LINDA-P is used as a probabilistic
baseline model. While LINDA-D deterministic nowcasts do not add any perturbations, LINDA-P does add them, as well as
BPS velocity perturbations with `lucaskanade/fmi+mch` parameters (Pulkkinen et al., 2019). Other parameters are set to
be data specific or to their default values.

The STEPS model is used in addition to LINDA-P as a probabilistic baseline. While being a bit older and having lower
discriminative power, it is a popular method for making reliable probabilistic precipitation nowcasts to this day. STEPS is
applied to dBZ reflectivity fields, also taking in the last three input images in addition to the motion field. Field perturbations
as well as motion field perturbations are applied with the same parameters as with LINDA-P. Six cascade levels are used for
the cascade decomposition, and the precipitation threshold of 8 dBZ is given as the lowest observable precipitation intensity.

## A3 Details on probabilistic verification metrics

### A3.1 Continuous Ranked Probability Score (CRPS)

The CRPS generalizes the Mean Absolute Error to probability distributions by calculating the sum of the difference between
the cumulative density function (CDF) of the nowcast and the empirical CDF of observations. It is defined as

$$\text{CRPS}(F, y) = \int_{-\infty}^{\infty} (F(\hat{y}) - \mathbb{1}(y \geq \hat{y}))^2 \mathrm{d}\hat{y}, \tag{A1}$$

where $\hat{y}$ denote possible forecast values, $F(\hat{y})$ the forecast CDF, and $\mathbb{1}(y \geq \hat{y})$ the empirical CDF of observations $y$.



### A3.2 Receiver Operating Characteristic (ROC) curve

The Receiver Operating Characteristic (ROC) curve (Mason, 1982; Wilks, 2011) quantifies the discriminative power of an ensemble for predicting over a certain threshold, by keeping track of the False Alarm Rate (FAR), i.e.,

$$\text{FAR} = \frac{\text{FP}}{\text{FP} + \text{TN}}, \tag{A2}$$

where the rate of false positives is indicated by FP, and the rate of true negatives is indicated by TN, against the the Probability of Detection (POD), i.e.,

$$\text{POD} = \frac{\text{TP}}{\text{TP} + \text{FN}}, \tag{A3}$$

where TP is the rate of true positives and FN is the rate of false negatives. POD is regularly binned, and FAR is averaged over those bins, making a curve, the area under which (AUC, Area Under the Curve) summarizes the overall discriminative power of the nowcasting method. An ROC AUC of 0.5 indicates zero skill, whereas a value of 1.0 indicates a perfect forecast. For ROC curve computations, we use 10 bins.

### A3.3 Reliability diagram

The reliability diagram (Wilks, 2011) measures the reliability of the forecast by presenting the observed relative frequencies of dBZ threshold exceedance events against the forecast probability of those events. Having these two values strongly correlate makes the forecast reliable. Reliability diagrams are built by dividing the forecast probabilities into bins (we choose 10), and incrementing them with associated binary indicators of whether the event happened. Sharpness histograms represent the number of events recorded in each forecast probability bin. They measure the relative "decisiveness" of the forecast, where a high decisiveness is associated with a convex histogram shape. A low decisiveness on the other hand can be discerned from a more uniform, or in the extreme case, a concave histogram shape.

### A3.4 Expected Calibration Error (ECE)

The Expected Calibration Error (ECE) (Naeini et al., 2015) quantitatively summarizes the reliability of a model indicated by a reliability diagram. It is defined as

$$\text{ECE} = \frac{1}{N} \sum_{b=1}^{B} n_b \mid f_b - o_b \mid, \tag{A4}$$

with a total of $N$ pairs of forecast probability and observation, forecast probabilities divided into $B$ bins, with $n_b$ observations per bin, $f_b$ mean bin forecast probability, and $o_b$ corresponding observation frequency in the bin. ECE corresponds to the MAE of the reliability diagram to the diagonal, weighted by the number of observations per bin.

### A3.5 Rank histogram

Rank histograms (Wilks, 2011) measure the bias and spread of ensemble nowcasts. They present a histogram of the rank of the true observed echo reflectivity among all ensemble members, where a convex histogram indicates a small spread and a concave



histogram indicates a small spread. On the other hand, a skew towards the left indicates a positive bias of predictions, and a
skew towards the right indicates a negative bias.

## A4 Details on deterministic verification metrics

### A4.1 Mean Error (ME)

The mean Error (ME) (Wilks, 2011) measures the bias of deterministic predictions. It is defined as

$$
\mathrm{ME} = \frac{1}{P} \sum_{p=1}^{P} \boldsymbol{y}_p - \hat{\boldsymbol{y}}_p. \tag{A5}
$$

for images or time series of them with $P$ pixels. This metric tells us about the mean bias of nowcasts produced. The absolute
value of ME is used to give a quantitative summary of the bias of predictions made.

### A4.2 Equitable Threat Score (ETS)

The Equitable Threat Score (ETS) (Hogan et al., 2010; Wilks, 2011) is an extension of the Threat Score, also known as the
critical success index (Schaefer, 1990). ETS aims to provide an estimate of deterministic skill in forecasting precipitation above
a certain intensity threshold. This extension takes into account the effect of randomly occurring true positives. ETS is defined
as

$$
\mathrm{ETS} = \frac{\mathrm{TP} - \mathrm{rnd}}{\mathrm{TP} + \mathrm{FN} + \mathrm{FP} - \mathrm{rnd}},
$$
$$
\text{where } \mathrm{rnd} = \frac{(\mathrm{TP} + \mathrm{FN})(\mathrm{TP} + \mathrm{FP})}{\mathrm{TP} + \mathrm{FN} + \mathrm{FP} + \mathrm{TN}}, \tag{A6}
$$

where the rnd term estimates the influence of random true positives.

### A4.3 Radially-Averaged Power Spectrum Density (RAPSD)

The Radially-Averaged Power Spectrum Density (RAPSD) (Ruzanski and Chandrasekar, 2011; Ulichney, 1988) measures how
well the power spectrum of precipitation is maintained, when calculated for nowcasts at different lead times. RAPSD fidelity
is summarized as

$$
\mathrm{RAPSD\ rel.\ MAE} = \frac{1}{F} \sum_{f=1}^{F} \frac{|\, P_f^{\mathrm{obs}} - P_f^{\mathrm{pred}} \,|}{P_f^{\mathrm{obs}}} \tag{A7}
$$

which is the absolute error between the observed and predicted PSD, relative to observed PSD, averaged over frequencies.
Here, $F$ denotes the number of frequencies of the power spectrum, $P_f^{\mathrm{obs}}$ the power of the observed field at the $f$:th frequency,
and $P_f^{\mathrm{pred}}$ the power of the predicted field at the $f$:th frequency. Taking the relative values allows comparing spectral densities
on multiple scales. In the present case, PSD frequencies are sampled linearly, weighting corresponding wavelengths towards
smaller scales, effectively biasing small-scale errors to be more important. This is however not necessarily a problem, as
prediction fidelity at small scales is the most important question we seek to answer with RAPSD.



## A5   Hardware and software packages used


The DEUCE model was built on PyTorch (version `1.12.1`). PyTorch Lightning (version `1.7.7`) was used to organize the neural network training and prediction workflow, and the TyXe library (version `0.0.1`) was used to turn DEUCE Bayesian, making use of the Pyro (version `1.4.0`) probabilistic programming language as its back-end for variational inference. The DEUCE training and prediction were performed using the Finnish IT Center for Science (CSC) supercomputer Puhti, using

one NVIDIA V100 GPU with 32GB of VRAM, 64GB of RAM, and 10 cores from a 2.1 GHz Intel Xeon Gold 6230 CPU. For the evaluation of the model performance, we used the pySTEPS library (version `1.6.1`). It served to produce baseline extrapolation-based model nowcasts, to calculate verification metrics, and to help with their visualization. The pySTEPS-based verification pipeline was run on a computational server of the Finnish Meteorological Institute, equipped with two Intel Xeon Gold 6138 2.0 GHz CPUs with each 20 cores and two threads by core, as well as with 192GB of RAM.

## Appendix B:  Additional figures

*Author contributions.*  BH performed the experiments and analysis, developed the methodology, wrote and maintained software, performed the validation as well as the visualization work, and wrote the original draft. SP acquired funding and resources, and in addition administrated the project. SP and TM jointly supervised BH in the work. All authors: BH, SP, and TM, took part in the conceptualization, as well as in the review & editing of the manuscript.

*Competing interests.*  The authors declare that they have no conflict of interest.

*Acknowledgements.*  We gratefully acknowledge the Finnish CSC — IT Center For Science for providing us with the computational resources used for the training and inference of the model developed, as well as V. Chandrasekar for his advice and critical comments regarding the work.



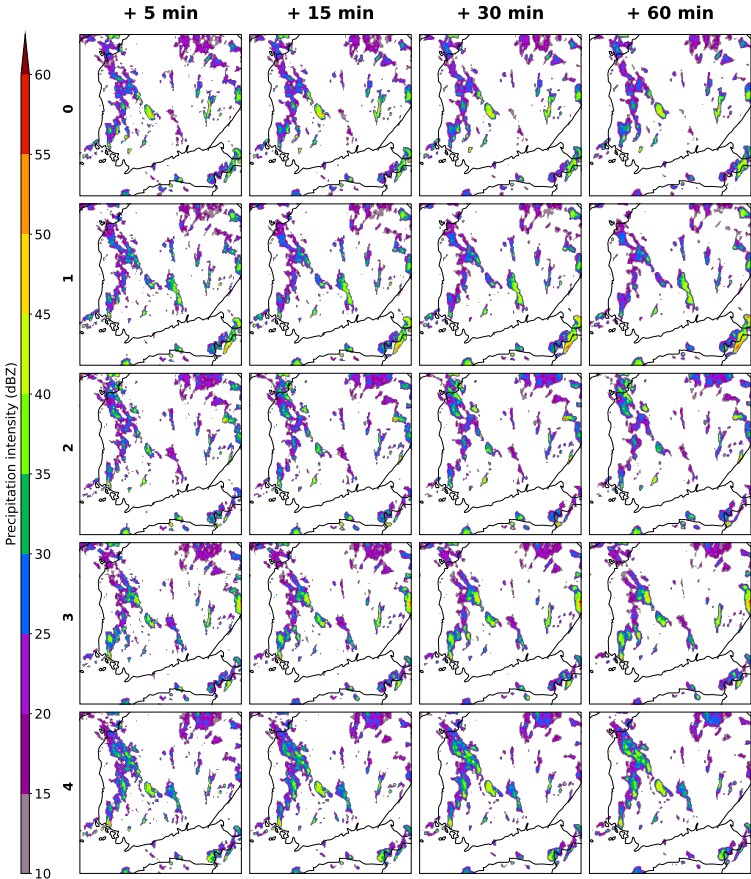

**Figure B1.** Five randomly selected examples of post-processed DEUCE ensemble members on the case studied, whose area covers southern Finland, starting at 15:00:00 UTC on 9 July 2022. The prediction lead times illustrated are 5, 15, 30, and 60 minutes.

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
