# Peer review of "DEUCE v1.0: A neural network for probabilistic precipitation nowcasting with aleatoric and epistemic uncertainties"

_EGUsphere, 2023_

## Author Response (AR1)

**Author Response**

**This author response consists of three distinct parts:**

1. Content and answer to the first reviewer comment (RC1)
2. Content and answer to the second reviewer comment (RC2)
3. The manuscript marked-up with changes made by the authors

**Parts 1. and 2. will consist of a sequence of:**

> The next reviewer comment cited.

The corresponding author answer in blue.

*The lines changed in the revised marked-up manuscript in italics, presented in order.*

**Reviewer comment 1**

> The authors consider a Bayesian neural network for precipitation nowcasting that is based on the U-Net architecture (DEUCE). This method esmtiates the total predictive uncertainty of precipition, subdividing the uncertainty into epistemic and aleatoric uncertainties. The model provides development scenarios up to 60 minutes. DEUCE is trained and evaluated for Finnish Meteoroloigcal Institute radar composites against established methods. First results seem to be a promising approach improving precipitation nowcasting.

Dear reviewer,
We thank you for the valuable comments that you provided us. We have addressed the points mentioned, and our response to each of them will be detailed below.

*-*

**General comments:**

> The text is written very well and the illustrations are very good, and helpful. The evaluation of the different uncertainties is valuable. The authors observe that most of the uncertainty is of aleatoric nature, and that the contribution of epistemic variance is universally. A challenging rainfall event is chosen as a case study. This is a good idea. But the reader might also be interested in how well the model performs at a more frequently occurring precipitation event.

The case study chosen at first indeed is a rather intense mostly convective rainfall event. Despite the quantitative verification representing the average performance of the model across a diverse corpus of events, we added a second case study, which concerns a rather different stratiform large-scale precipitation event. There, we found many of the same features

that were found with the first case study, and we believe that the addition of this case study gives a better general view of how the model performs in different scenarios to the reader.

*L316, Table 1, L327-L341, L395-L399, L403, L411, Figure 6, L422-L423, Figure 7, L427, Figure 8, Figure 9, L717-L735, Figure B1, Figure B2, Figure B3, Figure C1*

> The results are discussed carefully and in detail. There is some information about the data preprocessing missing. How is the input data distributed? This is important in order to understand the plausibility of the application of the used mathematical method.

To address the concerns about the data distribution, we added a histogram of the dataset reflectivity, highlighting the fact that those reflectivity values that are the most likely to represent precipitation are normally distributed, motivating the modeling method of the predictive distribution.

*Figure 4, L287-L294*

**Small comments:**

> Notation in 2.1 , first paragraph: Real value bold(y), do you mean real valued vector? Theta is not defined, is it also vector valued? Could you give a hint, what (kind of) parameter theta is?

We attempted to clarify the notation in subsection 2.1. What was meant by *real* was observed, or ground-truth. Next, the symbols **x**, **y**, **ŷ** were clarified to mean tensors, and **θ** a list of tensors . We hope that these changes will make the sentences concerned more clear and understandable. .

*L153-L154*

> Please explain with more detail: Line 190: How is D_KL defined?

We expanded the KL-divergence D_KL to its definition in Eq. 1.

*L191, Equation 1*

> Line 280: You normalize the data between zero and one. Maybe I missed it, but how is the input data distributed? Precipitation usually is skew symmetric, is the data transformed into a normally distribution? Pleas check, if it is mathematically correct to apply all methods to not-normally-distributed (?) data

It is correct to note that precipitation is not normally distributed (but log-normally). However, in this work we are applying the model to reflectivity and not precipitation rates. Radar reflectivity is known to follow a normal distribution because rain rate follows a log-normal distribution [1], and we verified whether this is in addition the case for our input data specifically with histograms of the data reflectivity of the dataset. We found that the part of the reflectivity distribution corresponding most likely to precipitation and not clutter or noise has indeed a Gaussian shape. Tangentially, we clarify the language used to make it clear that we are

approaching precipitation nowcasting through working with radar reflectivity values, which may further be used to make actual quantitative precipitation rate or accumulation predictions.

*L239-L243, L379, L381-L382, Figure 7, Figure 8, L483, Figure 15*

**Reviewer comment 2**

> Overall, the manuscript is well-written and the main results are clearly highlighted throughout the text. All the figures are appropriately labeled and capitoned; it's evident that the authors have devoted significant effort to effectively communicating their results. Consequently, I believe it absolutely deserves to be published in Geoscientific Model Development.
>
> However, I am flagging this manuscript as a minor revision because there are a couple of important areas (see major comments below) that deserve a more careful examination along with several minor grammatical and typographical errors. However, once these are addressed, I will be happy to enthusiastically recommend the revised manuscript for publication.

Dear reviewer,
We thank you for the valuable comments that you provided us. We have addressed the points mentioned, and our response to each of them will be detailed below.

-

**Major comments:**

> While the authors address both points below at some length in Section 5, I am still not fully satisfied with their given explanations from a machine learning perspective.
>
> 1. The inability of DEUCE to accurately forecast precipitation at smaller scales (see Fig. 15) deserves additional discussion. For instance, how does the DGMR research for Ravuri et al. (2021) address this issue? How do the authors plan to augment their current model to improve their model expressivity at small scales?
> 2. A secondary area of concern for the 1. d by training the model on weighted input samples with weights determined according to the observed precipitation distribution quantiles?
>
> A simple starting point could be to train the model for more epochs. From L285-286, I can infer that the current training procedure is only ~30 epochs. Given that the batch size is only N=2, this is lower than most neural networks of a similar size. Another potential area of improvement could be to use normalize the MSE in the first term of eq. 5 with the forecast output (see for example eq. 4 of https://arxiv.org/abs/2310.02994) to prevent the error from larger scales to dominate the loss.

We addressed the comments by adding discussion relating to the two facets mentioned from a machine learning perspective. Since what was asked is minor revisions, we did not include

any supplementary experiments in the manuscript. We still carried some experiments exploring the suggestions made, and our findings are presented below.

1. We tried training for more epochs (60 in total) but did not find that increasing the number of training epochs improves validation performance in the present case. However we cannot exclude the potential effect of external factors such as our learning rate schedule.
2. We tried to weight the likelihood part of the loss according to the inverse of the density of the pixel reflectivity value in the dataset distribution. Unfortunately, this did only result in considerable over-forecasting and oddly-behaving aleatoric uncertainties. Despite of this type of weighting being difficult to get right, we acknowledge the data imbalance problem in precipitation nowcasting giving rise to the importance of weighting samples such that all, even rare occurrences, can reasonably contribute to the gradients.

[Figure]

Illustration of the predictive mean and standard deviation of a base DEUCE checkpoint, a checkpoint with longer training, and a checkpoint trained with the loss weighting scheme described on a case taken from the prediction split (2019-05-25 13:00:00 UTC).

For this run, we slightly modified the training procedure as we found that the original DEUCE checkpoint sometimes experienced instabilities (NaN loss) in training for longer. The base DEUCE checkpoint here did not experience the same issues but had similar performance to the original one and was trained for 37 epochs with a batch size of 8, a sample size of 4, using 256x256px random crops from the 512x512px area, with learning rate decay after 5 epochs of non-improving validation loss (start at 1e-4). The "long" version started from the base checkpoint but was trained until 60 epochs. In the visualization, we show it after 57 epochs, as that was where its validation performance peaked. The "weighted" version was again trained from scratch with the same hyper-parameters except for the loss function. It achieved its peak validation performance after 28 epochs, which is the checkpoint shown in the figure.

The discussion section was modified such that the paragraph on small-scale variability was reworked, and a new paragraph was added next to it describing potential ways to mitigate the

underforecasting and lack of small-scale variability problems together from a machine learning perspective. We presented the two methods which you suggested together with some of our own ideas.

*L516-L552*

**Minor comments:**

> L7: Omit comma in "...deep learning methods, more capable..." and replace by "...deep learning methods which are more capable..."

This was fixed.

*L7*

> L75: Unclear what "discriminative deep learning models" refers to here since, as far as I understand, all the models described in the previous paragraph have at least some generative component to them.

To the extent of our knowledge, all of the models presented above L75 are discriminative, as they only model the conditional probability of predicted outputs given input frames, maximizing its likelihood through the MSE loss. In contrast, generative models such as GANs model the joint probability distribution of the outputs and inputs. We have however removed the word "discriminative" from the sentence such as to reduce possible confusion, because the same point can be made without its explicit usage.

*L75-L76*

> The post-processing procedure described in L240-252 for correctly approximating the spatio-temporal structure of ensemble members is quite impressive. I was wondering if the authors could add 1-2 lines in the Conclusion discussing how this step could be performed within a neural network setup.

After some thought, we indeed came by a possible method to better incorporate the correlated noise sampling procedure into the network, which was detailed in the conclusions section :

- "We could also think of directly appending the post-processing sampling with spatially correlated noise to the neural network, or even learning context-dependent spatiotemporal correlation structures. The sampled outputs could then be, e.g., fed to a GAN-like discriminator module, which would drive the processed outputs to be more realistic while retaining the uncertainty decomposition. "

*L603-L606*

> L324-325: Omit "the" in "...the 9 July 2022...;" add "upto" instead of "at" in "...leading at 15:00 UTC..."

We fixed that.

*L337, L340*

> L353: Omit "all" in "...which we all computed..."

This has been fixed too.

*L368*

> L421: Rephrase "...keeps open the possibility..."

"...keeps open the possibility for..." rephrased to "...means that we cannot exclude... "

*L437*

> L423: Rephrase "The suspicions are..."

Rephrased to "Such dependencies are..."

*L439*

> L426: Replace "to" with "for" in "...explanation to this..."

This has been fixed.

*L442*

> L431: Replace "some very" with "a" in "...some very slight increase..."

This has been fixed.

*L447-L448*

> L503: What does "variety" mean in "variety of the ensemble"?

We meant "breadth of the ensemble" and replaced "variety" here, acknowledging the ambiguity of the meaning of "variety" in this context.

*L520*

> L507-509: Rephrase the sentence: "Hence, despite ... results of Ravuri et al. (2021)." because it's unclear what the main point here is. See also major comment 1 above.

The paragraph containing was restructured in an attempt to separate and make clearer the different points made. See major comment modifications.

*L518-L536*

**The manuscript marked-up with changes made by the authors**

[revised manuscript text omitted]

---

## Author Response (AR2)

**Preamble**

**This author response will be structured as follows:**

There will be a succession of:

1. The editor review comment in question in a quotation block
2. The author response to that comment in blue
3. The possible changes made in the manuscript related to the comment in *italics*

for each of the editor review comments made. In addition, the manuscript marked up with changes will be appended in its entirety.
* * *
**Author Response**

Dear topic editor,

Thank you for your thoughtful comments and suggestions, which we have gone through. Our comments and revisions are detailed below.
* * *
> 1. For Figures 7, 8, and B2, please superposition the observation (isotherms of 15, 25, 35, and 45 dBZ) on the probability plots. This will help to compare the probability and observation.

Thank you for the suggestion. The comparison is indeed easier to make when overlaying the observation isotherms upon predicted exceedance probabilities.

- *We changed Figure 7, Figure 8, Figure B2, and Figure B3, as well as their captions.*
* * *
> 2. I suggest incorporating the findings associated with the second case in the discussion section. In particular, DEUCE lost the skill at small scales after 30 minutes and the skill in the area with substantial precipitation growth. It is also worth mentioning that LINDA-P provides a good reflectivity forecast on small scales, even with a smaller ensemble spread.

Thank you. We added a discussion on those aspects. Losing the skill at small scales after 30 minutes was linked to the original discussion on small scale variability. A new paragraph touching upon the lack of ability in predicting convective initiation was added to the discussion. The last comment was handled by shortly summarizing the pros and cons of STEPS and LINDA at the beginning of the discussion section.

- *L495-498: Adding the pros and cons of STEPS, LINDA, and mentioning that DEUCE offers a seemingly good compromise.*
- *L520-521: Mention that losing small-scale variability is esp. noticeable for lead times over 30 minutes.*
- *L554-558: Paragraph on the difficulty of predicting convective initiation.*
* * *
3. Line 156: Doesn't Lout correspond to the number of "output" timesteps? Is it a typo?

Thank you. This was indeed a typo, which has now been fixed.

- *Line 156: input -> output*
* * *
4. For clarification, does DEUCE provide the deterministic predictive precipitation (y_mean), and the ensemble is then determined by the breadth (sigma^2_pred, a combination of aleatoric and epistemic uncertainties)? Is the probability forecast (Fig. 7 and 8) determined more by epistemic uncertainties at the later forecast lead time and for the more vigorous intensity?

Thank you for the good questions. For the first one, the answer is *yes*, as DEUCE provides a predictive mean and variance, and the ensemble nowcast is determined by the predictive distribution parametrized as ~ N(y_mean, sigma^2_pred). However, we believe that it would be potentially misleading to call y_mean in its current form deterministic, as its values still depend on the weights drawn. An explanation for the process of using DEUCE to generate ensemble nowcasts is given in Sect. 2.3.

As for the second question about whether the probability forecast of the first case is determined more by epistemic or aleatoric uncertainty, it helps to take a look at Figures 9 and 10 as well. Figure 9 shows the composition of the standard deviation for case 1. There, we see that because the square of the aleatoric standard deviation is much larger than that that of the epistemic standard deviation, the predictive standard deviation consists almost exclusively of aleatoric uncertainty ( as sigma_pred = sqrt(sigma_ep ^2 + sigma_al^2). The figure below, showing the proportion of both components in the predictive variance, helps illustrating the same effect. From this figure, it also becomes clear that the contribution of epistemic uncertainty actually decreases with lead time in this case.

[Figure]

The proportion of epistemic and aleatoric variance inside of the predictive variance. Grey areas indicate pixels where these proportions could not be calculated due to the variance summing to zero.

Figure 10 shows that the same trend can be observed across diverse circumstances, with solely the proportion of aleatoric uncertainty increasing with lead time (top right chart). Finally, we show here in the figure below how the effect is seen in exceedance probabilities. Only utilizing aleatoric uncertainty yields exceedance probabilities very close to predictive uncertainty, but epistemic uncertainty by itself looks very different, with narrower distributions, close to predictive means (The effect of prediction smoothing can be clearly seen, as smaller features are not predicted well by the mean anymore).

[Figure]

25dBZ exceedance probabilities for the first case using predictive SD, separate aleatoric SD, and separate epistemic SD.

- *L431-432: Added a sentence about aleatoric uncertainty overshadowing epistemic uncertainty in probabilistic nowcasts, because of their overlapping spatial extent.*
* * *
5. The central innovative part of the methodology is considering aleatoric and epistemic uncertainties for probabilistic forecasts. However, given that DEUCE still has the issue of losing small-scale skill, can the authors briefly comment on how such limitation is linked to aleatoric and epistemic uncertainties in the conclusion section?

Thank you for the comment. Although the relationship between the smoothing of predictions and uncertainties was already touched upon in the discussion section (L517-527), a small paragraph further discussing the subject from the point of view proposed was added in conclusions. It was discussed that not accounting for any increase in prediction skill, sharper predictions would likely increase epistemic uncertainty but not affect aleatoric uncertainty, except for circumstances where excessive smoothing is the mechanism limiting the prediction of high enough reflectivity values.

- *L600-606: Added a small paragraph on the effect of smoothing on aleatoric and epistemic uncertainties.*
* * *
> 6.The predictive variance in DEUCE determines the ensemble distribution (Eq. 7). This differs from the spread definition in the traditional NWP-based ensemble forecast used for the probabilistic forecast. This may be worth to be mentioned or emphasized in the introduction.

Thank you for the suggestion. Changes in wording have been made in the abstract and introduction to make it clearer that with DEUCE, ensembles follow from the estimated predictive distribution and not the other way around. Also, the contrast with NWP is highlighted in Sect. 2.3, where the generation of ensemble nowcasts is described.

- *L12 : and produces -> using them to produce*
- *L129-130: Change in the wording to make it clear that the predictive distribution leads to the ensemble nowcast. Also, removal of out-of-place explanation of the role of radar nowcasting w.r.t. precipitation nowcasting (already present at L39-41).*
- *L242-245: Mentioning the difference between the DEUCE ensemble definition just presented above, and the NWP-based ensemble definition.*
* * *
**The manuscript marked-up with changes made by the authors**

[revised manuscript text omitted]